# Learning-Time Encoding Shapes Unlearning in LLMs

**Ruihan Wu**[*]  **Konstantin Garov**[*]    **Kamalika Chaudhuri**
Department of Computer Science
University of California, San Diego
`{ruw076,kgarov,kamalika}@ucsd.edu`

## Abstract

As large language models (LLMs) are increasingly deployed in the real world, the ability to "unlearn", or remove specific pieces of knowledge post hoc, has become essential for a variety of reasons ranging from privacy regulations to correcting outdated or harmful content. Prior work has proposed unlearning benchmarks and algorithms, and has typically assumed that the training process and the target model are fixed. In this work, we empirically investigate how learning-time encoding in knowledge encoding impact the effectiveness of unlearning factual knowledge. We conduct two studies: (i) examining how paraphrased descriptions influence unlearning performance, and (ii) analyzing unlearning when multiple facts are embedded within the same training text chunk. Our empirical study reveals two important implications: a new perspective for interpreting unlearning performance and practical strategies for improving LLM unlearning.

## 1 Introduction

Large Language Models (LLMs) acquire vast amounts of factual knowledge through large-scale pretraining as well as subsequent fine-tuning. As they are increasingly deployed in real applications, there is an increasing need for "unlearning" certain information in an efficient post-hoc way (Bourtoule et al., 2021; Liu et al., 2025) from pre-trained or the fine-tuned models. This need arises for several reasons. One is compliance with privacy regulations such as the GDPR's "Right to be Forgotten" (gdp, 2016) – for example, when a user requests that personal data used during training be removed. Other use cases include addressing copyright violations (Eldan & Russinovich, 2023; Dou et al., 2024; Vyas et al., 2023), removing unsafe or harmful content (such as instructions for building weapons) (Yao et al., 2024b; Li et al., 2024), and removing personal and sensitive information (Jang et al., 2022; Wu et al., 2023; Barrett et al., 2023). These diverse scenarios often align with slightly different objectives for the unlearning process.

One common goal of unlearning in LLMs is to make specific factual knowledge non-extractable, which means that prevent the model from generating it in response to relevant prompts (Jang et al., 2022; Si et al., 2023; Guo et al., 2024; Tian et al., 2024; Choi et al., 2024; Yuan et al., 2025; Wu et al., 2024; Patil et al.), and at the same time retain the remaining knowledge. Prior work has primarily focused on benchmarks (Maini et al.; Shi et al., 2024; Yao et al., 2024a; Jin et al., 2024) and developing algorithms (Ilharco et al., 2022; Si et al., 2023; Zhang et al.; Yu et al., 2023; Wu et al., 2023; Jia et al., 2025; Eldan & Russinovich, 2023; Patil et al.), and typically assume that both the trained model and the unlearning targets are fixed. The central goal in these studies is to improve the effectiveness of the unlearning method itself.

However, a crucial factor is often overlooked: the way a model is trained – including how knowledge is encoded in the training data – may significantly influence how challenging it is to later unlearn that knowledge. Existing work has only partially addressed this dimension: Zhao et al. (2024) examine training-related factors for data unlearning, which differs from knowledge unlearning in LLMs, while Krishnan et al. (2025) focus narrowly on the frequency of target knowledge in the training set. To our knowledge, no prior work has systematically studied how training-time knowledge encoding shapes

---

[*]Equal contribution

the unlearning process in LLMs. In this paper, we take a step toward filling this gap by addressing the fundamental question:

**How does learning-time knowledge encoding affect knowledge unlearning in LLMs?**

To ensure fair comparison, we investigate this question through controlled experiments. For this purpose, we extend two existing unlearning datasets – Eval-DU (Wu et al., 2024) and TOFU (Maini et al.) – resulting in *Eval-DU+* and *TOFU+*. Both datasets involve synthetic biographies of fictional characters that are highly unlikely to occur in the pre-training corpus; this allows us to control the knowledge space and the exact textual encodings observed by the LLM during training. We fine-tune two LLMs (Llama2-7B and Gemma2-2B) on identical sets of factual knowledge, varying only the knowledge textual encoding. After fine-tuning, we attempt to unlearn specific pieces of knowledge and analyze the differences in the unlearning across different types of encoding. Notably, our study focuses on unlearning from fine-tuned models, a common scenario where sensitive content or private user data could be introduced[1].

Using the constructed testbed, we first empirically study the effect of paraphrased texts on knowledge unlearning. Two seemingly conflicting intuitions motivate this study. On the one hand, training on multiple paraphrased descriptions of a knowledge piece may strengthen its memorization, thereby making the piece harder to erase. On the other hand, prior work (Allen-Zhu & Li, 2024) suggests that paraphrased training data encourage models to internalize knowledge in a more structured manner, which could in turn make unlearning easier—particularly when the unlearning request is phrased differently from the original training text. Thus, it remains unclear whether augmenting knowledge with paraphrased encodings in the training corpus ultimately helps or hinders unlearning. Our empirical results reveal two key findings:

1. Unlearning is more difficult when the knowledge pieces targeted for unlearning (forget set) were encoded with multiple paraphrases in the training data. Conversely, unlearning is more efficient when the knowledge pieces not targeted for unlearning (retain set) were paraphrased during training.

2. When both the forget and retain sets were represented with multiple paraphrased descriptions, overall unlearning effectiveness improves.

Second, we aim to empirically investigate unlearning when training units are multi-fact text chunks. This setting reflects more realistic cases: in natural corpora, knowledge is rarely presented in isolation but embedded within longer passages—such as Wikipedia paragraphs—that intertwine multiple facts. In practice, unlearning requests may apply only to a subset of the facts in a chunk, while the rest must be preserved. Our empirical study yields three key findings:

1. Unlearning individual knowledge pieces becomes significantly difficult when forget and retain facts are entangled within the same chunk.

2. Unlearning is relatively more effective when the forget set aligns with chunk boundaries in the training data.

3. Unlearning individual facts is easier when they are at least isolated from retain knowledge within the same chunk (e.g., expressed in separate sentences).

Finally, we discuss two implications from our empirical findings. First, our empirical study provides a new angle to interpret the unlearning performance. Some applicable scenarios are *surprising algorithmic failures*, *variance across benchmarks*, and *variance across models*. Second, our empirical results suggest two potential strategies to improve the post-hoc efficiency of unlearning for large language models: *paraphrasing*, that is using multiple paraphrased descriptions of knowledge during fine-tuning, and *separating*, that is structuring the training data to avoid text entanglement along potential unlearn and retain splits in the future unlearning.

---

[1] We also include experiments with causal language modeling, same as the pre-training objective, and multiple LLM architectures, which may offer indirect evidence toward generalization to pretrained models. However, due to the lack of visibility into pretraining data of the existing publicly pre-trained models and limited computational resources for pretraining from scratch on a sufficiently large controlled corpus, we leave formal validation of this generalization to future work.

## 2 PROBLEM SET-UP

Intuitively, the behavior of unlearning, as an invertion of learning, should be shaped by learning-time choices such as learning algorithms (k-nearest neighbour or parametric learning) or model architectures (linear models or deep models). Prior work (Allen-Zhu & Li, 2024; Allen-Zhu & Li) suggests that in the context of large language model (LLM), the knowledge encodings used in the training data are one of the most important factors in LLM knowledge acquisition. This, raises a natural question: *How does the behavior of unlearning a piece of knowledge $k$ vary depending on how $k$ was encoded during training?* In this paper, we investigate two concrete study settings to address this question.

### 2.1 SETTING I: THE EFFECT OF TEXT PARAPHRASING ON UNLEARNING

**Target paraphrasing and unlearning difficulty.** Prior work (Zhao et al., 2024) claims that a deeper memorization of training data might make unlearning harder. By extending this claim to the context of knowledge memorization in LLM, we hypothesize that when the knowledge is encoded in the training corpus through multiple paraphrased descriptions, this knowledge piece is harder to be erased – an unlearning algorithm must suppress all of them, which increases the difficulty compared to removing a single unique description.

To empirically validate this hypothesis, we propose the following testing framework and state the problem after. Given a knowledge piece $k$ we consider two modes of encoding it: 1) as a single text: $\{t_0^k\}$ and 2) as three different paraphrased texts: $\{t_1^k, t_2^k, t_3^k\}$. For a fixed knowledge space $K$ and a subset $K_{ul} \subset K$ targeted by an unlearning algorithm, this gives rise to three modes of training datasets $D_{train}$, based on the encoding mode used on the forget set $K_{ul}$ and on the retain set $K \setminus K_{ul}$:

1. `FT-Single`: all knowledge pieces are encoded with single texts or $D_{train} = \bigcup_{k \in K} \{t_0^k\}$

2. `FT-Unlearn-Mul`: forget knowledge pieces are encoded with multiple texts while retain knowledge pieces only with single texts or: $D_{train} = \bigcup_{k \in K_{ul}} \{t_1^k, t_2^k, t_3^k\} \cup \bigcup_{k \in K \setminus K_{ul}} \{t_0^k\}$

3. `FT-Retain-Mul`: forget knowledge pieces are encoded with single texts while retain knowledge pieces only with multiple texts (conversely to FT-Mix) or: $D_{train} = \bigcup_{k \in K_{ul}} \{t_0^k\} \cup \bigcup_{k \in K \setminus K_{ul}} \{t_1^k, t_2^k, t_3^k\}$

**Problem 1** *Among the three models trained on three modes of training data* `FT-Single`, `FT-Unlearn-Mul` *and* `FT-Retain-Mul` *respectively, for which training data mode is unlearning the forget set $K_{ul}$ the most difficult?*

If the intuition holds, we expect the relative difficulty to follow the order: `FT-Unlearn-Mul` > `FT-Single` > `FT-Retain-Mul`

**Training corpus paraphrasing and unlearning effectiveness.** Prior studies (Allen-Zhu & Li) have shown that paraphrasing training data can lead LLMs to internalize knowledge in a more structured manner and the knowledge is more extractable through different formats of prompts. A plausible implication is that such structured representations also make it easier for unlearning algorithms to target and remove specific knowledge: if the model has organized a concept systematically, then unlearning could proceed more directly and effectively.

To test this hypothesis, we compare two training regimes: `FT-Single`, where each knowledge piece is encoded by a single text, and `FT-Mul`, where each knowledge piece is encoded by three paraphrased texts ($D_{train} = \bigcup_{k \in K} \{t_1^k, t_2^k, t_3^k\}$). We then ask:

**Problem 2** *Between models trained on* `FT-Single` *and* `FT-Mul`, *which presents a greater challenge for unlearning the forget set $K_{ul}$.*

If the intuition is correct, unlearning should be more effective for the model trained on `FT-Mul`, since its structured knowledge representations may allow the algorithm to target $K_{ul}$ more systematically. *Importantly, this question is not resolved by Problem 1*: paraphrasing only the forget set might increase difficulty, and paraphrasing only the retain set might reduce it. When both are paraphrased, these effects may offset one another, leaving the net impact uncertain.

```
┌─────────────────────────────────┐ ┌──────────────────────────────────────────────────────────┐
│ Examples of Eval-DU+ Dataset    │ │ Examples of TOFU+ Dataset                                  │
│                                 │ │                                                            │
│ The textual description for     │ │ The textual description for knowledge k:                   │
│ knowledge k:                    │ │ Q: Who is this celebrated LGBTQ+ author from Santiago,     │
│ Reid Perry has Richard Perry    │ │ Chile known for their true crime genre work?               │
│ as his father.                  │ │ A: The author in question is Jaime Vasquez, an esteemed    │
│                                 │ │ LGBTQ+ writer who hails from Santiago, Chile and           │
│ The paraphrased description for │ │ specializes in the true crime genre.                       │
│ knowledge k:                    │ │                                                            │
│ The father of Reid Perry is     │ │ The paraphrased description for knowledge k:               │
│ Richard Perry.                  │ │ Q: Could you tell me about the celebrated LGBTQ+ author    │
│                                 │ │ from Santiago, Chile who excels in the true crime genre?   │
│ The text trunk that describes   │ │ A: Jaime Vasquez is the celebrated author recognized       │
│ multiple knowledge pieces       │ │ within the LGBTQ+ community and beyond for their           │
│ including k:                    │ │ exceptional work in true crime, hailing from Santiago,     │
│ Richard Perry, born in 1956 in  │ │ Chile                                                      │
│ Maryland, works as an airline   │ │                                                            │
│ pilot. He is married to Parker  │ │ The text trunk that describes multiple knowledge pieces    │
│ Ross and is the father of       │ │ including k:                                               │
│ Reid, Reed, Raymond, and        │ │ Q: Who is Jaime Vasquez, and what is notable about his     │
│ Quentin Perry. Richard's        │ │ contributions to literature?                               │
│ parents are…                    │ │ A: Jaime Vasquez is a celebrated LGBTQ+ author from        │
│                                 │ │ Santiago, Chile, born on February 25, 1958. With a         │
│                                 │ │ father ... he channels his passion for storytelling        │
│                                 │ │ into the true crime genre. His award-winning books,        │
│                                 │ │ including ...                                              │
└─────────────────────────────────┘ └──────────────────────────────────────────────────────────┘
```

Figure 1: Examples of different textual descriptions in two datasets Eval-DU+ and TOFU+.

## 2.2 SETTING II: THE UNLEARNING FROM TEXT CHUNKS

In natural datasets, knowledge is rarely presented in isolation; instead, it is embedded within longer passages that intertwine multiple facts. For example, a biography of a public figure may simultaneously describe personal details and professional accomplishments. Such entanglement introduces interactions between the structure of the data and the unlearning task. In this work, we aim to investigate how unlearning behaves under these different forms of interaction.

**Individual fact unlearning within chunks.** In practice, the unlearning incentive may apply to only one knowledge item within a paragraph, while the remaining items should be preserved. For instance, in a biography, personal details (e.g., birth date, address) may need to be unlearned, while professional achievements should remain intact. When such information is expressed in the same textual context—often with overlapping wording and intertwined descriptions (see example in Figure 1)—removing only the sensitive portion could be exceptionally challenging.

To study this setting, we introduce a new training mode, `FT-Mul-Chunk`. The training corpus consists of paraphrased paragraphs: $D_{train} = \cup_{i=1}^{I}\{p_1^i, p_2^i, p_3^i\}$, where each set $\{p_1^i, p_2^i, p_3^i\}$ contains paraphrases of the same paragraph. Each paragraph $p_j^i$ encodes a set of knowledge pieces $K_i \subset K$, with the full knowledge space partitioned as $K = \cup_{i=1}^{I} K_i$. We define the unlearning target $K_{ul}^{ind} \subset K$ such that each paragraph $K_i$ contributes only one or a few knowledge items to this target set. We ask the following problem:

**Problem 3** *Given a model trained under* `FT-Mul-Chunk`*, how difficult is it to unlearn the target subset* $K_{ul}^{ind}$*, where each element is entangled within a larger paragraph that also encodes retain knowledge?*

**Chunk-aligned unlearning.** A realistic scenario is when all knowledge pieces contained in a text chunk must be removed together—for example, an individual may request the deletion of their entire personal record from a model. Under the `FT-Mul-Chunk` training mode, we define the unlearning target as $K_{ul}^{align} = \cup_{i \in I_{ul}} K_i$, i.e., the union of entire paragraph-level knowledge sets. Intuitively, since the forget set and retain set do not co-occur within the same text chunk $p_j^i$, the trade-off between unlearning and retention may be easier to preserve.

**Problem 4** *Is unlearning the target* $K_{ul}^{align}$ *easier than unlearning the more granular target* $K_{ul}^{ind}$*?*

**Isolated unlearning within chunks.** If unlearning becomes easier when forget and retain knowledge appear in different chunks, a natural follow-up is whether *intra-chunk isolation*, where each knowledge piece is described in a separate sentence, further facilitates unlearning.

To test this, we define an additional training mode, `FT-Mul-Chunk-Iso`. Here the corpus is $D_{train} = \cup_{i=1}^{I}\{p_1^i, p_2^i, p_3^i\}$ where each paragraph $p_j^i$ encodes knowledge set $K_i$ such that each knowledge piece is written in its own sentence; here is an example

> ```
> Parker Ross is the wife of Richard Perry.  As a child, Reed Perry
> belongs to Richard Perry.  Poppy Perry is Richard Perry's aunt...
> ```

$p_2^i, p_3^i$ are paraphrased versions at the sentence level (still keeping the intra-isolation). As in Problem 3, the unlearning target is $K_{ul}^{ind}$.

**Problem 5** *Does unlearning the target $K_{ul}^{ind}$ become less challenging when the model is trained under* `FT-Mul-Chunk-Iso` *compared to* `FT-Mul-Chunk`*?*

The study for Problems 3–5 will collectively highlight a new and non-obvious insight: the structure and lexical organization of the training text, not merely chunk-level co-occurrence, plays a crucial role in whether unlearning succeeds.

### 2.3 RATIONALE FOR FOCUSING ON FINE-TUNING.

Our study focuses on unlearning from fine-tuned models, an important use-case in which sensitive or private user data is often introduced during customization for downstream tasks. It also allows precise control over the knowledge space. While the study targets fine-tuning, we include causal language modeling (the same training objective as pre-training) and multiple LLM architectures, which may offer indirect evidence toward generalization to pretrained models. A formal investigation of the novel unlearning problem proposed in this work within the pre-training setting remains an important direction. However, we leave this to future work due to limited transparency in data of the existing pre-trained models and the high computational cost of pretraining a model from scratch on a sufficiently large and controlled corpus.

## 3 EXPERIMENTAL SET-UP

**Unlearning set-up.** We experiment with two representative unlearning algorithms that are also evaluated in previous benchmarks (Maini et al.; Shi et al., 2024; Wu et al., 2024):

1) **gradient ascent (GA)** (Jang et al., 2022) which removes knowledge by ascending the loss on the unlearning dataset. The strength of unlearning is controlled by the number of ascending steps $t$.

2) **task vector (TV)** (Ilharco et al., 2022; Zhang et al., 2023) computes the parameter difference vector between the original model $\theta_{\text{original}}$ and a model $\theta_{\text{overfit}}$ trained to overfit the unlearning data. The final model is then defined as $\theta_{\text{unlearn}} = \theta_{\text{original}} - \alpha(\theta_{\text{overfit}} - \theta_{\text{original}})$, where the scaling factor $\alpha$ controls the strength of unlearning.

In the unlearning dataset, we include 1 or 3 unseen during training textual descriptions for unlearning each knowledge piece $k$ in the forget set. We denote the choices of two unlearning algorithms and two unlearning texts as `GA-Single`, `GA-Mul`, `TV-Single` and `TV-Mul` respectively. In the Appendix G.2, we additionally experimented with **Gradient Difference** that explicitly leverages retain data for five proposed problems; the results share the same observations as TV and GA.

**Evaluation.** Similarly to existing unlearning benchmarks (Maini et al.; Shi et al., 2024; Wu et al., 2024), we evaluate the unlearning effectiveness through the trade-off between forgetting the target knowledge and retaining the non-target (retain) knowledge. To evaluate the degree to which a model "knows" a knowledge space $K$ we compute the average probability of correctly completing the unseen (during fine-tuning and unlearning) description of the knowledge pieces in $K$. In Appendix E, we provide more details alongside evaluation via completing the descriptions used during fine-tuning and via average QA accuracy.

To quantify the unlearn-retain trade-off, we vary the parameter controlling the trade-off (e.g. $t$ in GA and $\alpha$ in TV). For each parameter value we obtain a model checkpoint, whose unlearn and retain scores we compute. These scores are plotted to form a trade-off curve, where curves closer to the top-left indicate a more favorable trade-off. We then compute normalized area-under-curve **Norm-AUC ($\uparrow$)** (to account for different initial scores) for these curves. Please check the details of metrics in Appendix E as well as the full curves of the experiments in Appendix F. We incorporate an additional absolute evaluation metric as a complements: Retain@Unlearn $\tau$. This metric asks:

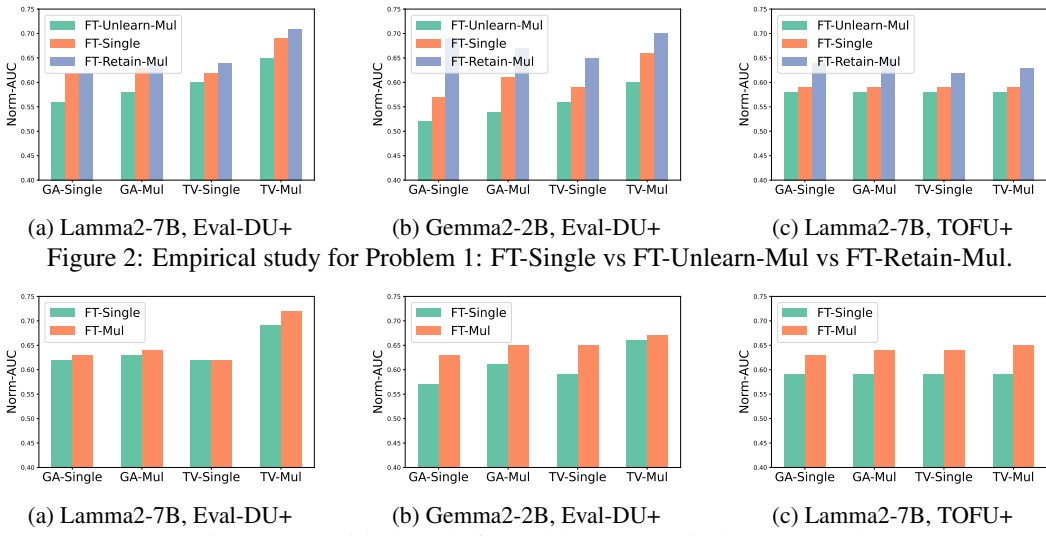

Figure 2: Empirical study for Problem 1: FT-Single vs FT-Unlearn-Mul vs FT-Retain-Mul.

Figure 3: Empirical study for Problem 2: FT-Single vs FT-Mul.

when the forgotten knowledge is suppressed to a small target level $\tau$. We presented the results for this metric in Appendix G.3, which share the similar obsertations.

**Datasets construction.** In order to systematically study how learning-time knowledge encodings affect unlearning, we augment two existing unlearning datasets — Eval-DU (Wu et al., 2024) and TOFU (Maini et al.) — to form **Eval-DU+** and **TOFU+**. We reuse their knowledge spaces: Eval-DU consists of 862 biographical or family relationships facts involving 100 fictitious individuals where each fact is a knowledge piece and TOFU contains 200 fictitious authors with 20 QA pairs per author where each QA defines a knowledge piece. We then augment both datasets by: (1) multiple paraphrased descriptions for each individual knowledge piece, and (2) multiple paraphrased text chunks for each designed partition of the knowledge set. Figure 1 shows the data examples in Eval-DU+ and TOFU+. **Eval-DU+ and TOFU+ allow the experiments across these two knowledge spaces and two text formats (narrative texts and QAs)**, which serves a robust testbed for analyzing how learning-time knowledge encodings influence the unlearning. For more details about the dataset construction and the unlearn-retain split, please check the details in Appendix D.

**Models and fine-tuning.** Our experiments involve three large language models: Llama2-7B (Touvron et al., 2023), Gemma2-2B (Team et al., 2024), and Qwen3-4B (Yang et al., 2025). We evaluate five combinations of models and datasets: (Llama2-7B, Eval-DU+), (Gemma2-2B, Eval-DU+), and (Llama2-7B, TOFU+), (Qwen3-4B, Eval-DU+), (Qwen3-4B, TOFU+). We expect our findings to remain consistent across two datasets and multiple model families, supporting broader generalization to unseen models and datasets. The results of Qwen3-4B will be presented in Appendix G.1.

Fine-tuning procedures all start from the public pre-trained models. For Eval-DU+, we perform fine-tuning with Causal Language-Modeling (same objective as the pre-training (Radford et al., 2018)), which minimizes the next-token prediction loss over all tokens in each training example. In contrast, as TOFU+ is structured in a QA format, we adopt supervised fine-tuning (Radford et al., 2018; Ouyang et al., 2022): each QA pair is placed in a predefined QA template, and the objective is to minimize the loss only over the answer tokens. We use the Adam optimizer for all fine-tuning experiments and update all model parameters during fine-tuning. Please check more implementation details as well as the fine-tuning results in Appendix F.

## 4 EXPERIMENT RESULTS

In this section, we empirically investigate the problems defined in Section 2. Since our focus is on how textual knowledge encodings in the training data affect downstream unlearning, each study follows the same procedure: we train LLMs under selected training modes (defined in Section 2), apply a fixed unlearning algorithm to the resulting models, and then evaluate unlearning performance. Experiments are conducted across two knowledge spaces (Eval-DU+ and TOFU+), three pre-trained

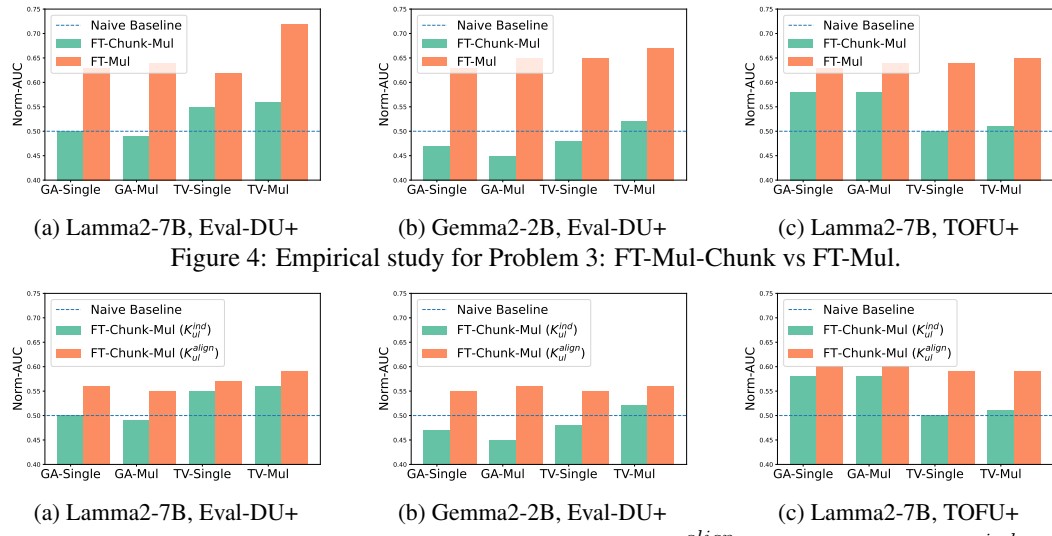

(a) Lamma2-7B, Eval-DU+  (b) Gemma2-2B, Eval-DU+  (c) Lamma2-7B, TOFU+

Figure 4: Empirical study for Problem 3: FT-Mul-Chunk vs FT-Mul.

(a) Lamma2-7B, Eval-DU+  (b) Gemma2-2B, Eval-DU+  (c) Lamma2-7B, TOFU+

Figure 5: Empirical study for Problem 4: FT-Mul-Chunk ($K_{ul}^{align}$) vs FT-Mul-Chunk ($K_{ul}^{ind}$).

LLMs (Llama2-7B, Gemma2-2B, and Qwen-4B), and two unlearning algorithms (GA and TV). We visualize the results in this section and present the original numbers in Table 8 in the appendix.

## 4.1 EMPIRICAL STUDY I: THE EFFECT OF TEXT PARAPHRASING ON UNLEARNING

In this section, we empirically study how paraphrased descriptions in the training dataset affect the difficulty of unlearning (Problem 1 and Problem 2).

**Empirical study for Problem 1.** To test whether having multiple paraphrased descriptions of the same knowledge piece makes it harder to remove, we compare unlearning performance across three training modes: `FT-Single`, `FT-Unlearn-Mul`, and `FT-Retain-Mul`. Figure 2 reports the results. We find that models trained with `FT-Unlearn-Mul` exhibit consistently worse unlearning performance compared to `FT-Single`, while models trained with `FT-Retain-Mul` perform consistently better. From these results, we can now answer Problem 1: **unlearning is most difficult under the `FT-Unlearn-Mul` regime.** The results suggest that difficulty depends on where paraphrasing occurs: 1. when knowledge pieces in the forget set $K_{ul}$ were presented with multiple paraphrased versions during, unlearning becomes harder; conversely, when knowledge pieces in the retain set are paraphrased, unlearning is more effective, since the model can better preserve non-targeted knowledge while removing the target set.

**Empirical study for Problem 2.** We next study whether paraphrasing the entire training corpus makes downstream unlearning more effective. To this end, we compare `FT-Single` against `FT-Mul`, where every knowledge piece is encoded with multiple paraphrases. Figure 3 presents the results. Models trained with `FT-Mul` consistently achieve better unlearning performance than those trained with `FT-Single`. We can therefore answer Problem 2: **models trained on paraphrased corpora (`FT-Mul`) exhibit more effective unlearning.** This provides empirical suggesting that the hypothesis that training on paraphrased descriptions encourages LLMs to internalize knowledge in a more structured manner (Allen-Zhu & Li, 2024), which benefits the subsequent unlearning process.

**Takeaway.** Together, these results highlight that paraphrasing influences unlearning: paraphrasing the forget set makes unlearning harder and symetrically paraphrasing the retain set makes it easier, and more interestingly, paraphrasing the entire corpus improves overall unlearning effectiveness.

## 4.2 EMPIRICAL STUDY II: THE UNLEARNING FROM TEXT CHUNKS

In this section, we examine the task of unlearning knowledge embedded within larger text chunks (Problem 3, Problem 4, and Problem 5).

**Empirical study of Problem 3.** We first evaluate the challenge of unlearning individual facts when their descriptions are entangled with other knowledge within larger chunks. To this end, we train

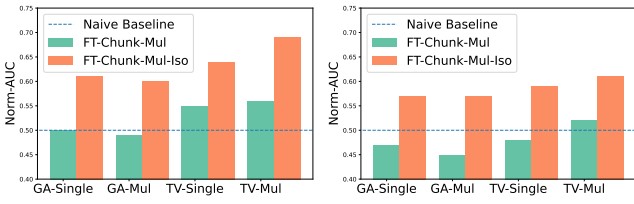

(a) Lamma2-7B, Eval-DU+      (b) Gemma2-2B, Eval-DU+

Figure 6: Empirical study for Problem 5: FT-Mul-Chunk vs FT-Mul-Chunk-ISO.

models under the `FT-Mul-Chunk` regime and measure unlearning performance. As shown in Figure 4, we find that, except for the case of GA with Llama2-7B on TOFU+, unlearning is almost entirely ineffective. A Norm-AUC value near 0.5 indicates that the unlearning algorithm removes both target and retained knowledge at similar rates; we plot the value of $0.5$ in the dashed line as the baseline. This stands in sharp contrast to the results when each training sample encodes a single knowledge piece such as `FT-Mul`, where AUC values are generally around or above 0.6, despite using the same knowledge space $K$ and unlearning split $K_{ul}$.

We can now answer Problem 3: **unlearning individual facts from text chunks is exceptionally challenging.** We hypothesize that this difficulty arises because the learning dynamics of target and retain knowledge are strongly correlated due to the entangled wording, making them hard to separate. Supporting evidence comes from Allen-Zhu & Li (2024), who show that when models are trained with paraphrased paragraphs about a set of knowledge pieces, for example a paragraph of biography, then a single entity embedding (e.g., a person's name in the 'biography' example) can internally encode all associated facts. More supporting evidence is provided in Zhao et al. (2024), which also claims that unlearning is harder when the retain and forget sets are more entangled. The entanglement of the textual description in our case is a specification in the LLM data.

**Empirical study of Problem 4.** Next, we examine unlearning when the forget set aligns with chunk boundaries. Specifically, we target $K_{ul}^{align}$, where entire chunks are to be removed. Figure 5 shows that unlearning performance on $K_{ul}^{align}$ is consistently better than on the more granular $K_{ul}^{ind}$. We can now answer Problem 4: **unlearning chunk-aligned targets ($K_{ul}^{align}$) is more effective than unlearning granular targets ($K_{ul}^{ind}$).** The likely explanation is similar to Problem 3: when forget and retain knowledge are less correlated (i.e., they do not co-occur within the same chunk), the unlearn–retain trade-off is easier to maintain.

**Empirical study of Problem 5.** Finally, we study whether unlearning becomes easier when individual facts are isolated inside a chunk. We train models with `FT-Mul-Chunk-ISO`, where each knowledge piece is expressed in a separate sentence. As shown in Figure 6, we observe that unlearning individual targets $K_{ul}^{ind}$ is substantially more effective under `FT-Mul-Chunk-ISO` compared to `FT-Mul-Chunk`. We can now answer Problem 5: **unlearning from `FT-Mul-Chunk-ISO` is more effective than from `FT-Mul-Chunk` when targeting same individual facts.** This reinforces our earlier hypothesis: isolating knowledge reduces correlation between forget and retain sets within the same chunk, making the unlearn–retain trade-off easier to preserve.

**Takeaway.** Across all three problems, our results show that text chunk structure plays a decisive role in unlearning: when forget and retain knowledge are entangled within the same passage, unlearning individual facts is extremely difficult; aligning the forget set with chunk boundaries makes unlearning more effective; and further isolating individual facts within chunks provides the greatest gains.

## 5 DISCUSSION

We discuss several implications of our empirical findings. First, our results provide a new perspective on interpreting unlearning performance that is orthogonal to algorithmic choices. In particular, they help explain discrepancies that arise in specific scenarios, such as the following:

1. **Surprising algorithmic failures.** If we find the unlearning algorithms all fail – AUC$\approx 0.5$. Our results suggest this may not be due to algorithm weakness alone, but to the fact that the forget and retain sets are entangled in the same training text chunk.

2. **Variance across benchmarks.** Suppose an unlearning algorithm performs better on one than the other. Without our lens, one might conclude that the second benchmark is just "harder." With our results, we can explain why some benchmark is harder from the aspects of training data and the unlearn split. This shifts interpretation from "algorithmic deficiency" to "dataset structure effect."

3. **Variance across models.** Suppose we evaluate the same unlearning algorithm across different pre-trained models on a shared benchmark. One might observe that the algorithm performs better on one model than the other. Our work sheds light on this discrepancy: it can arise from differences in the models' pre-training corpora, even when both corpora might cover the same knowledge space.

Second, our findings point to potential learning-time strategies for improving the post-hoc efficiency of unlearning in large language models.

1. **Paraphrasing.** Introducing multiple paraphrased descriptions of knowledge during fine-tuning appears to lead to more structured internal representations, which in turn make later unlearning more effective as well. Notably, Allen-Zhu & Li (2024) suggest that paraphrasing training data facilitates knowledge extraction; our work complements this by proposing that paraphrasing also enhances unlearning effectiveness.

2. **Separating.** Structuring training data so that knowledge likely to be subject to future unlearning requests is disentangled from retain knowledge[2]. This design reduces the correlation between forget and retain sets and enables a cleaner unlearn–retain trade-off.

More broadly, these results highlight that unlearning is not solely a problem of algorithm design, but also of representation and data curation. Future work could explore how to deliberately structure or augment training corpora to make future unlearning easier, and whether similar principles hold in multimodal or cross-lingual settings.

## 6 RELATED WORK

**Machine unlearning and training data.** The most relevant research to ours is Zhao et al. (2024), which study machine unlearning at the data level and show that forget sets with higher memorization or stronger entanglement with the retain set are more difficult to unlearn. While these observations resonate with our findings at a high level, our work focuses on knowledge unlearning in the context of LLMs. In particular, we analyze in detail how different training corpus designs influence memorization of knowledge and how specific textual descriptions create varying degrees of entanglement between forget and retain sets. Fan et al. (2024) study worst-case forget sets in the context of data unlearning, while our work focuses on knowledge unlearning in large language models. Moreover, rather than analyzing only which forget set is chosen, we investigate how the format of the overall training corpus, including paraphrasing and chunk structure, affects the difficulty of unlearning. In parallel, Krishnan et al. (2025) examine how the frequency of a knowledge piece in the training corpus affects unlearning difficulty[3]. Our work considers this factor as well (Problem 1), but extends beyond their scope by studying frequency effects not only in the forget set, but also in the retain set and across the entire training corpus.

For more related work about algorithms and evaluations in machine unlearning for LLMs, we discuss them in details in Appendix C

## 7 CONCLUSION AND FUTURE WORK

**Conclusion.** In summary, this work takes an initial step toward understanding how learning-time knowledge encoding influences post-hoc unlearning in large language models. Through controlled ex-

---

[2]This may seem paradoxical: if the unlearning target is known in advance, why not remove it before training? However, unlearning requests often arise after deployment, particularly when training data is collected from public sources. For instance, some celebrities may not want LLMs to retain family-related information from their Wikipedia pages, while others may prefer that it be preserved. Such preferences are difficult to anticipate at training time.

[3]Our work is independently conducted and concurrent with Krishnan et al. (2025)

periments, we show that how the training text paraphrasing can ainfluence forgetting and retention and how chunk structure determines whether individual facts can be removed effectively. Together, these findings offer a new perspective for interpreting unlearning outcomes across models, benchmarks, and algorithms, and suggest practical strategies to improve the post-hoc efficiency of unlearning.

**Limitations and future work.** Although this paper focuses on the role of training data choices in unlearning, several other learning-time factors may also influence unlearning effectiveness. These include the model architecture (e.g., full-parameter tuning LoRA (Hu et al., 2022)) and the learning algorithm (e.g., supervised fine-tuning vs. reinforcement learning (Rafailov et al., 2023; Lu et al., 2022)). A promising direction for future work is to systematically investigate how such factors impact the behavior and difficulty of unlearning. Another meaningful direction is to understand the mechanism behind the findings in this paper: how do the difference encodings translate into unlearning performance? Thirdly, due to limited computational resources, our experiments are restricted to LLMs that undergo fine-tuning. While we believe the findings presented in this paper may generalize to the pretraining stage and to unlearning from pretrained models directly, validating this hypothesis remains an important avenue for future research when more resources are available.

## REPRODUCIBILITY STATEMENT

We provide detailed dataset construction and implementation information in Appendix D and Appendix E. In addition, at the beginning of the Appendix, we include an anonymous link that enables reproduction of our main experimental results.

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

## A ORGANIZATION OF THE APPENDIX

The organization of this appendix is as below:

1. In Section B, we discuss the usage of LLMs in this work.

2. In Section C, we discuss some additional related work in the direction of LLM unlearning.

3. In Section D, we present the details of constructing benchmark datasets Eval-DU+ and TOFU+, including the detailed statistics of paraphrasing, the templates for generating the synthetic texts, and an illustration of calculating the knowledge score *prbability* in Eval-DU+.

4. In Section E, we will present the implementation details in our experiments, including the compute resources used in the epxeriments, the details of model fine-tuning, and the details of the unlearning.

5. In Section F, we will present additional experimental results, including the performance of fine-tuned models on LLM general benchmarks, the full unlearning results, and the full plots of trade-off curves used for calculating the Norm-AUC.

Our code for reproducing the results is anonymously released at `https://anonymous.4open.science/r/knowledge_encoding_for_llm_unlearning-62BB/README.md`.

## B THE USE OF LLMS

We employ large language models (LLMs) primarily to improve the grammar and clarity of our writing. In addition, the synthetic datasets used in our controlled experiments are constructed by prompting LLMs. All research ideas, directions, and decisions, however, are independently conceived and carried out by the authors.

## C ADDITIONAL RELATED WORK

**Machine unlearning for LLMs: algorithms.** Recently, machine unlearning for LLMS has emerged as an important area of research Liu et al. (2025); Si et al. (2023). In this work, we focus on GA Jang et al. (2022); Barbulescu & Triantafillou (2024) and TV (task vector) Ilharco et al. (2022) methods. Other notable approaches include: NPO Zhang et al.; Bronec & Helcl (2025) which utilizes the DPO objective Rafailov et al. (2023) treating the unlearn data as negative preference data, WHP uses a linear combination of the distributions induced by initial and a reinforced model as an unlearn model Eldan & Russinovich (2023); Liu et al. (2024b), UWC calibrates the post-unlearning parameters with the initial parameters to better preserve the model's utility Wang et al. (2024a), GRU uses both the unlearning and retention gradients at each update step Wang et al. (2024a). Regularizers are often employed to better preserve the model's utility. For example: augementing the unlearning objective with the retention gradient (GDR) Maini et al.; Zhang et al.; Liu et al. (2022) and regularizing with the KL divergence on the retention set (KLR) Maini et al.; Zhang et al.. Non-training based methods include: localization-informed unlearning Li et al. (2024); Meng et al. (2022); Wu et al. (2023) which localize the components of the LLM related to the forget data and black-box in-context unlearning Pawelczyk et al. (2023). Other recent promising approaches are Jia et al. (2024); Liu et al. (2024a); Ji et al. (2024); Wang et al. (2024b); Ishibashi & Shimodaira (2023); Thaker et al. (2024b); Wang et al. (2025); He et al. (2025).

**Machine unlearning for LLMs: evaluations.** Evaluating the effectiveness machine unlearning method poses another challenge. As an example, Eldan & Russinovich (2023) uses completion and question-answer probability-based scores, while Lynch et al. (2024) proposes comparing the unlearned model and a model retrained on the retention data. UNCD uses Cognitive Diagnosis Modeling for fine-grained evaluation Lang et al. (2025). Besides TOFU (Maini et al.) and Eval-DU (Wu et al. (2024)), several other benchmarks have been proposed to assess the effectiveness of unlearning in LLMs such as: WMDP - a dataset consisting of hazardous knowledge in multiple-choice format Li et al. (2024) and RWKU for zero-shot konwledge unlearning Jin et al. (2024), MUSE proposes a comprehensive benchmark evaluating six desirable properties from the perspectives of both data owners and model deployers Shi et al. (2024), and PEBench for multimodal LLMs Xu et al.

(2025). Finally, Thaker et al. (2024a) discusses the limitations of existing benchmarks. Beyond this it shows that entanglement of retain and unlearn data in test prompts decreases the evaluation score of an unlearned model.

# D   DETAILS OF CONSTRUCTING BENCHMARK DATASETS

**Detailed statistics of paraphrasing.**   We present the statistics of the paraphrasing and how they are used for training, unlearning and evaluation in both datasets Eval-DU+ and TOFU+:

| Dataset | # paraphrasing for each $k$ | | | # paraphrasing of text chunks |
|---------|----------|-----------|------------|-------------------------------|
|         | Training | Unlearning | Evaluation |                               |
| Eval-DU+ | 3 | 3 | 3 | 3 |
| TOFU+ | 3 | 3 | 1 | 3 |

**Templates for the prompt when generating the texts through ChatGPT-4o.**   Here are the templates of how we generate the paraphrased descriptions for each knowledge piece given the initial texts provided by each original dataset and the paraphrased text chunks for each group of knowledge.

---

**Templates of generating the paraphrased descriptions for each knowledge piece**

**Eval-DU+**
```
Could you help rephrase the sentence {Initial Text} while
keeping the word {Objective Word}?  Please give me 8
variations.
```

**TOFU+**
```
Could you help rephrase both the question and the answer
below? Question:  {Intial Question}
Answer:  {Intial Answer}
Please give me 7 variations and list them as a sequence of
QAs, formated by 1., 2., ..., 7.
```

---

---

**Templates of generating the paraphrased text chunks for each knowledge group**

**Eval-DU+**
```
Here are the family information and biographic information
for {Person Name}.  Could you summarize all information in
one paragraph and give me 5 versions of them by shuffling the
order of these information:
{Text Description of the 1st Knowledge Piece}
...
Please list the versions by 1., 2., ...
```

**TOFU+**
```
Could you help summarize all information in the following 20
question-answering into one question-answer pair?
1.
Question:  {1st Question}
Answer:  {1st Answer}
...
Please give me 3 variations and do not miss any information.
Please response in the format
Variation 1:
Question 1:...
Answer 1:...
...
```

---

After collecting the responses from ChatGPT-4o, we did some text extractions in order to get a organized list of target paraphrased texts.

**Calculating knowledge scores in Eval-DU+ and TOFU+.**   In TOFU+, where $x_k$ is a QA pair, we adopt the "Probability" metric from the original TOFU benchmark: given a question embedded in a prompt template, the score is the likelihood the LLM assigns to generating the reference answer. In Eval-DU+, each $x_k$ is a narrative sentence. Notice that each knowledge piece has the structure tuple of (s, r, o). We are able to identify the keywords for s, r, or o in a given text description. For example, here is a text description for (*Richard Perry*, *father*, *Reid Perry*) and we highlight the corresponding keywords.

*Reid Perry* has *Richard Perry* as his *father*.

Then, we can calculate the likelihood of the keyword appearing the last in this sentence, which is *father*, for a given LLM which modelizes the likelihood function $\pi_\theta$.

**The definition of the text chunk in Eval-DU+ and TOFU+.**   The knowledge space of Eval-DU+ is partitioned by the subjects (person) in the factual tuple. The knowledge space of TOFU+ is partitioned by the fictitious authors. We then synthetically generate the text chunk for each partition the details are presented in the above paragraph.

**Unlearn–retain split in Eval-DU+ and TOFU+.**   In Eval-DU+, we construct the unlearn set $K_{ul}$ (and the corresponding $K_{ul}^{ind}$) by randomly selecting 100 out of 862 knowledge pieces, with the remaining pieces forming the retain set. For the setting of chunk-aligned unlearning, $K_{ul}^{align}$ is defined as all facts associated with 10 randomly chosen fictitious people.

In TOFU+, the retain set consists of 400 QAs randomly sampled from the first 198 authors. To construct $K_{ul}$ (and $K_{ul}^{ind}$), we randomly select 40 QAs from the same pool of authors. For $K_{ul}^{align}$, we adopt the original unlearn–retain split of the TOFU dataset, which contains 40 knowledge pieces associated with 2 out of the 200 authors.

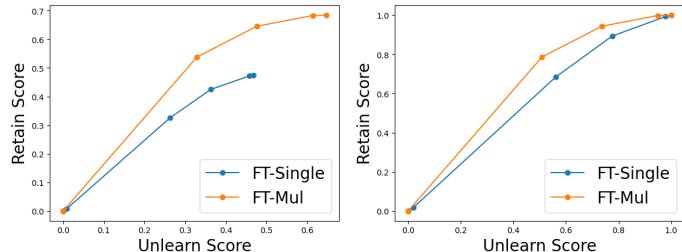

Figure 7: Illustrations for Norm-AUC. Left: Raw (unnormalized) trade-off curves. Right: Normalized curves used to compute Norm-AUC. All plots show the unlearning with GA. The dataset is `Eval-DU+` and the target `Llama2-7B` models are fine-tuned with either `FT-Single` or `FT-Mul`.

# E    ADDITIONAL DETAILS IN EXPERIMENTS

**Compute resources in the experiment.**    All experiments are conducted by NVIDIA RTX 6000 Ada GPU. Each run of the fine-tuning and the unlearning is run on two GPUs. The fine-tuning will take 6-12 hours, and each run of the unlearning process as well as the evaluation will take will take 4-8 hours; the time varies on different models.

**Quantitative metrics for evaluating the trade-off: Norm-AUC and AUC.**    To evaluate the unlearn-retain trade-off for an unlearning method, we vary the parameter controlling the trade-off (e.g. $t$ in GA and $\alpha$ in TV) across a list of pre-defined values. For each parameter value we obtain a model checkpoint, whose unlearn and retain scores we compute. These scores are plotted to form a trade-off curve (Figure 7), where curves closer to the top-left indicate a more favorable trade-off.

When comparing different fine-tuning strategies under a fixed unlearning configuration (i.e., using the same unlearning data and algorithm), the trade-off curves may start at different points due to the different fine-tuned models. For instance, models fine-tuned with `FT-Mul` typically achieve higher initial knowledge scores. To account for this we define the **Norm-AUC** ($\uparrow$). This metric first normalizes all knowledge scores by their value in the original fine-tuned model and then computes the area under the normalized curve (Figure 7, middle). A higher Norm-AUC indicates a more efficient unlearning and a Norm-AUC of 0.5 implies that unlearn and retain scores are decreasing at the same rate.

**Fine-tuning details.**    The batch sizes are 16 for all models fine-tuned on Eval-DU+ and 32 for the model fine-tuned on TOFU+. In addition, we pick the learning rate $\eta \in \{2 \cdot 10^{-5}, 10^{-5}, 2 \cdot 10^{-6}\}$ and the number of epochs $N \in \{1, \cdots, 8\}$ to ensure a good fit on the fine-tuning set while having a good test performance. The final selection of the two parameters are presented in Table 1.

Table 1: Hyperparameter values of the fine-tuning on different models and datasets: the learning rate $\eta$ and the number of epochs $N$

|  | Llama2-7B, Eval-DU+ | | Gemma2-2B, Eval-DU+ | | Llama2-7B, TOFU+ | |
| --- | --- | --- | --- | --- | --- | --- |
|  | $\eta$ | $N$ | $\eta$ | $N$ | $\eta$ | $N$ |
| FT-Single | $10^{-5}$ | 5 | $10^{-5}$ | 8 | $10^{-5}$ | 5 |
| FT-Unlearn-Mul | $10^{-5}$ | 5 | $10^{-5}$ | 8 | $10^{-5}$ | 5 |
| FT-Retain-Mul | $10^{-5}$ | 5 | $10^{-5}$ | 8 | $10^{-5}$ | 5 |
| FT-Mul | $10^{-5}$ | 5 | $10^{-5}$ | 8 | $10^{-5}$ | 5 |
| FT-Mul-Chunk | $10^{-5}$ | 4 | $10^{-5}$ | 8 | $10^{-5}$ | 4 |
| FT-Mul-Chunk-ISO | $10^{-5}$ | 4 | $10^{-5}$ | 8 | $10^{-5}$ | 4 |

**Unlearning details.**    We present the hyperparameter details for ach unlearning algorithm: gradient ascent (GA) has a list of step numbers $t$ to control the trade-off and the learning rate $\eta_{ga}$ (the batch sizes are fixed as 8 for Eval-DU+ and 16 for TOFU+), task vector (TV) has a list of scaling parameter values $\alpha$ to control the trade-off, as well as the number of epoch $T_{tv}$ and the learning rate $\eta_{tv}$ to

train the reinforced model (the batch sizes are fixed as 16 for Eval-DU+ and 32 for TOFU+). The values are picked to best present the trade-off. Their values given different fine-tuning data choices are presented as below:

Table 2: Hyperparameter values of `GA-Single`.

| | Llama2-7B, Eval-DU+ | | Gemma2-2B, Eval-DU+ | | Llama2-7B, TOFU+ | |
|---|---|---|---|---|---|---|
| | List of $t$ | $\eta_{ga}$ | List of $t$ | $\eta_{ga}$ | List of $t$ | $\eta_{ga}$ |
| FT-Single | $\{0, 5, 10, \cdots, 75\}$ | $3 \times 10^{-6}$ | $\{0, 5, 10, \cdots, 75\}$ | $3 \times 10^{-6}$ | $\{0, 5, 10, \cdots, 75\}$ | $3 \times 10^{-6}$ |
| FT-Mul | $\{0, 5, 10, \cdots, 75\}$ | $3 \times 10^{-6}$ | $\{0, 5, 10, \cdots, 75\}$ | $3 \times 10^{-6}$ | $\{0, 5, 10, \cdots, 75\}$ | $3 \times 10^{-6}$ |
| FT-Unlearn-Mul | $\{0, 5, 10, \cdots, 75\}$ | $3 \times 10^{-6}$ | $\{0, 5, 10, \cdots, 75\}$ | $3 \times 10^{-6}$ | $\{0, 5, 10, \cdots, 75\}$ | $3 \times 10^{-6}$ |
| FT-Retain-Mul | $\{0, 5, 10, \cdots, 75\}$ | $3 \times 10^{-6}$ | $\{0, 5, 10, \cdots, 75\}$ | $3 \times 10^{-6}$ | $\{0, 5, 10, \cdots, 75\}$ | $3 \times 10^{-6}$ |
| FT-Mul-Chunk | $\{0, 5, 10, \cdots, 75\}$ | $3 \times 10^{-6}$ | $\{0, 5, 10, \cdots, 75\}$ | $3 \times 10^{-6}$ | $\{0, 5, 10, \cdots, 75\}$ | $10^{-6}$ |
| FT-Mul-Chunk-ISO | $\{0, 5, 10, \cdots, 75\}$ | $3 \times 10^{-6}$ | $\{0, 5, 10, \cdots, 75\}$ | $3 \times 10^{-6}$ | $\{0, 5, 10, \cdots, 75\}$ | $10^{-6}$ |

Table 3: Hyperparameter values of `GA-Mul`.

| | Llama2-7B, Eval-DU+ | | Gemma2-2B, Eval-DU+ | | Llama2-7B, TOFU+ | |
|---|---|---|---|---|---|---|
| | List of $t$ | $\eta_{ga}$ | List of $t$ | $\eta_{ga}$ | List of $t$ | $\eta_{ga}$ |
| FT-Single | $\{0, 5, 10, \cdots, 75\}$ | $3 \times 10^{-6}$ | $\{0, 5, 10, \cdots, 75\}$ | $3 \times 10^{-6}$ | $\{0, 5, 10, \cdots, 75\}$ | $3 \times 10^{-6}$ |
| FT-Mul | $\{0, 5, 10, \cdots, 75\}$ | $3 \times 10^{-6}$ | $\{0, 5, 10, \cdots, 75\}$ | $3 \times 10^{-6}$ | $\{0, 5, 10, \cdots, 75\}$ | $3 \times 10^{-6}$ |
| FT-Unlearn-Mul | $\{0, 5, 10, \cdots, 75\}$ | $3 \times 10^{-6}$ | $\{0, 5, 10, \cdots, 75\}$ | $3 \times 10^{-6}$ | $\{0, 5, 10, \cdots, 75\}$ | $3 \times 10^{-6}$ |
| FT-Retain-Mul | $\{0, 5, 10, \cdots, 75\}$ | $3 \times 10^{-6}$ | $\{0, 5, 10, \cdots, 75\}$ | $3 \times 10^{-6}$ | $\{0, 5, 10, \cdots, 75\}$ | $3 \times 10^{-6}$ |
| FT-Mul-Chunk | $\{0, 5, 10, \cdots, 75\}$ | $3 \times 10^{-6}$ | $\{0, 5, 10, \cdots, 75\}$ | $3 \times 10^{-6}$ | $\{0, 5, 10, \cdots, 75\}$ | $10^{-6}$ |
| FT-Mul-Chunk-ISO | $\{0, 5, 10, \cdots, 75\}$ | $3 \times 10^{-6}$ | $\{0, 5, 10, \cdots, 75\}$ | $3 \times 10^{-6}$ | $\{0, 5, 10, \cdots, 75\}$ | $10^{-6}$ |

Table 4: Hyperparameter values of `TV-Single`.

| | Llama2-7B, Eval-DU+ | | | Gemma2-2B, Eval-DU+ | | | Llama2-7B, TOFU+ | | |
|---|---|---|---|---|---|---|---|---|---|
| | List of $\alpha$ | $N_{tv}$ | $\eta_{tv}$ | List of $\alpha$ | $N_{tv}$ | $\eta_{tv}$ | List of $\alpha$ | $N_{tv}$ | $\eta_{tv}$ |
| FT-Single | $\{0, 0.2, 0.5, 1.0, 5.0, 10.0\}$ | 20 | $10^{-5}$ | $\{0, 0.2, 0.5, 1.0, 5.0, 10.0\}$ | 20 | $10^{-5}$ | $\{0, 0.05, 0.1, 0.2, 0.3, 0.4, 0.5, 1.0, 5.0, 10.0\}$ | 20 | $10^{-5}$ |
| FT-Mul | $\{0, 0.2, 0.5, 1.0, 5.0, 10.0\}$ | 20 | $10^{-5}$ | $\{0, 0.2, 0.5, 1.0, 5.0, 10.0\}$ | 20 | $10^{-5}$ | $\{0, 0.05, 0.1, 0.2, 0.3, 0.4, 0.5, 1.0, 5.0, 10.0\}$ | 20 | $10^{-5}$ |
| FT-Unlearn-Mul | $\{0, 0.2, 0.5, 1.0, 5.0, 10.0\}$ | 20 | $10^{-5}$ | $\{0, 0.2, 0.5, 1.0, 5.0, 10.0\}$ | 20 | $10^{-5}$ | $\{0, 0.05, 0.1, 0.2, 0.3, 0.4, 0.5, 1.0, 5.0, 10.0\}$ | 20 | $10^{-5}$ |
| FT-Retain-Mul | $\{0, 0.2, 0.5, 1.0, 5.0, 10.0\}$ | 20 | $10^{-5}$ | $\{0, 0.2, 0.5, 1.0, 5.0, 10.0\}$ | 20 | $10^{-5}$ | $\{0, 0.05, 0.1, 0.2, 0.3, 0.4, 0.5, 1.0, 5.0, 10.0\}$ | 20 | $10^{-5}$ |
| FT-Mul-Chunk | $\{0, 0.2, 0.5, 1.0, 5.0, 10.0\}$ | 20 | $10^{-5}$ | $\{0, 0.2, 0.5, 1.0, 5.0, 10.0\}$ | 20 | $10^{-5}$ | $\{0, 0.2, 0.5, 1.0, 5.0, 10.0, 20.0, 30.0, 50.0\}$ | 400 | $10^{-5}$ |
| FT-Mul-Chunk-ISO | $\{0, 0.2, 0.5, 1.0, 5.0, 10.0\}$ | 20 | $10^{-5}$ | $\{0, 0.2, 0.5, 1.0, 5.0, 10.0\}$ | 20 | $10^{-5}$ | $\{0, 0.2, 0.5, 1.0, 5.0, 10.0, 20.0, 30.0, 50.0\}$ | 400 | $10^{-5}$ |

Table 5: Hyperparameter values of `TV-Mul`.

| | Llama2-7B, Eval-DU+ | | | Gemma2-2B, Eval-DU+ | | | Llama2-7B, TOFU+ | | |
|---|---|---|---|---|---|---|---|---|---|
| | List of $\alpha$ | $N_{tv}$ | $\eta_{tv}$ | List of $\alpha$ | $N_{tv}$ | $\eta_{tv}$ | List of $\alpha$ | $N_{tv}$ | $\eta_{tv}$ |
| FT-Single | $\{0, 0.2, 0.5, 1.0, 5.0, 10.0\}$ | 20 | $10^{-5}$ | $\{0, 0.2, 0.5, 1.0, 5.0, 10.0\}$ | 20 | $10^{-5}$ | $\{0, 0.05, 0.1, 0.2, 0.3, 0.4, 0.5, 1.0, 5.0, 10.0\}$ | 20 | $10^{-5}$ |
| FT-Unlearn-Mul | $\{0, 0.2, 0.5, 1.0, 5.0, 10.0\}$ | 20 | $10^{-5}$ | $\{0, 0.2, 0.5, 1.0, 5.0, 10.0\}$ | 20 | $10^{-5}$ | $\{0, 0.05, 0.1, 0.2, 0.3, 0.4, 0.5, 1.0, 5.0, 10.0\}$ | 20 | $10^{-5}$ |
| FT-Retain-Mul | $\{0, 0.2, 0.5, 1.0, 5.0, 10.0\}$ | 20 | $10^{-5}$ | $\{0, 0.2, 0.5, 1.0, 5.0, 10.0\}$ | 20 | $10^{-5}$ | $\{0, 0.05, 0.1, 0.2, 0.3, 0.4, 0.5, 1.0, 5.0, 10.0\}$ | 20 | $10^{-5}$ |
| FT-Mul | $\{0, 0.2, 0.5, 1.0, 5.0, 10.0\}$ | 20 | $10^{-5}$ | $\{0, 0.2, 0.5, 1.0, 5.0, 10.0\}$ | 20 | $10^{-5}$ | $\{0, 0.05, 0.1, 0.2, 0.3, 0.4, 0.5, 1.0, 5.0, 10.0\}$ | 20 | $10^{-5}$ |
| FT-Mul-Chunk | $\{0, 0.2, 0.5, 1.0, 5.0, 10.0\}$ | 20 | $10^{-5}$ | $\{0, 0.2, 0.5, 1.0, 5.0, 10.0\}$ | 20 | $10^{-5}$ | $\{0, 0.2, 0.5, 1.0, 5.0, 10.0, 20.0, 30.0, 50.0\}$ | 400 | $10^{-5}$ |
| FT-Mul-Chunk-ISO | $\{0, 0.2, 0.5, 1.0, 5.0, 10.0\}$ | 20 | $10^{-5}$ | $\{0, 0.2, 0.5, 1.0, 5.0, 10.0\}$ | 20 | $10^{-5}$ | $\{0, 0.2, 0.5, 1.0, 5.0, 10.0, 20.0, 30.0, 50.0\}$ | 400 | $10^{-5}$ |

## F  ADDITIONAL RESULTS

**Performance of fine-tuned models.**  We first ensure that each model achieves a near-perfect fit on its fine-tuning data – Table 6 shows the probabilities among fine-tuning set or the unseen test set. We additionally evaluate general utility on three standard LLM benchmarks: *MMLU* (Hendrycks et al., 2021) for multi-domain language understanding, *PIQA* (Bisk et al., 2020) for commonsense reasoning, and *RACE* (Lai et al., 2017) for reading comprehension. The results are presented in Table 7. We observe that fine-tuning does not significantly degrade performance on these general tasks, confirming that the models retain broad capabilities.

**Full tables of all unlearning results.**  We summarize all unlearning results in Table 8 for completeness and easier comparison and results reproducing for the future work.

**Full plots of trade-off curves.**  For completion, we attach the full trade-off curves for calculating Norm-AUC.

Table 6: Average knowledge scores among finetuning set (FT Probs.) or unseen test set (Test Probs.).

| | Llama2-7B, Eval-DU+ | | Gemma2-2B, Eval-DU+ | | Llama2-7B, TOFU+ | |
|---|---|---|---|---|---|---|
| | FT Probs. | Test Probs. | FT Probs. | Test Probs. | FT Probs. | Test Probs. |
| FT-Single | 0.95 | 0.47 | 0.97 | 0.39 | 0.99 | 0.12 |
| FT-Mul | 0.92 | 0.68 | 0.95 | 0.61 | 0.99 | 0.16 |

Table 7: Pretrained and finetuned LLMs on three general utility benchmarks.

| LLM & Dataset Metric | Llama2-7B on Eval-DU+ | | | Gemma2-2B on Eval-DU+ | | | Llama2-7B on TOFU+ | | |
|---|---|---|---|---|---|---|---|---|---|
| | MMLU | PIQA | RACE | MMLU | PIQA | RACE | MMLU | PIQA | RACE |
| Pre-train | 0.400 | 0.778 | 0.396 | 0.496 | 0.791 | 0.373 | 0.400 | 0.778 | 0.396 |
| FT-Single | 0.383 | 0.775 | 0.398 | 0.496 | 0.798 | 0.380 | 0.335 | 0.758 | 0.398 |
| FT-Mul | 0.368 | 0.782 | 0.392 | 0.486 | 0.792 | 0.365 | 0.332 | 0.773 | 0.402 |

Table 8: This summarize the Norm-AUC of all unlearning results at differnet setting of unlearning across two datasets and two models.

| Model, Dataset | FT Choices | Gradient Ascent | | Task Vector | |
|---|---|---|---|---|---|
| | | GA-Single | GA-Mul | TV-Single | TV-Mul |
| Llama2-7B, Eval-DU+ | FT-Single | 0.62 | 0.63 | 0.62 | 0.69 |
| | FT-Unlearn-Mul | 0.56 | 0.58 | 0.60 | 0.65 |
| | FT-Retain-Mul | 0.62 | 0.64 | 0.64 | 0.71 |
| | FT-Mul | 0.63 | 0.64 | 0.62 | 0.72 |
| | FT-Chunk-Mul | 0.50 | 0.49 | 0.55 | 0.56 |
| | FT-Chunk-Mul (align) | 0.56 | 0.55 | 0.57 | 0.59 |
| | FT-Chunk-Mul-ISO | 0.61 | 0.60 | 0.64 | 0.69 |
| Gemma2-2B, Eval-DU+ | FT-Single | 0.57 | 0.61 | 0.59 | 0.66 |
| | FT-Unlearn-Mul | 0.52 | 0.54 | 0.56 | 0.60 |
| | FT-Retain-Mul | 0.69 | 0.67 | 0.65 | 0.70 |
| | FT-Mul | 0.63 | 0.65 | 0.65 | 0.67 |
| | FT-Chunk-Mul | 0.47 | 0.45 | 0.48 | 0.52 |
| | FT-Chunk-Mul (align) | 0.55 | 0.56 | 0.55 | 0.56 |
| | FT-Chunk-Mul-ISO | 0.57 | 0.57 | 0.59 | 0.61 |
| Llama2-7B, TOFU+ | FT-Single | 0.59 | 0.59 | 0.59 | 0.59 |
| | FT-Unlearn-Mul | 0.58 | 0.58 | 0.58 | 0.58 |
| | FT-Retain-Mul | 0.64 | 0.65 | 0.62 | 0.63 |
| | FT-Mul | 0.63 | 0.64 | 0.64 | 0.65 |
| | FT-Chunk-Mul | 0.58 | 0.58 | 0.50 | 0.51 |
| | FT-Chunk-Mul (align) | 0.60 | 0.60 | 0.59 | 0.59 |

For the results in Section 4.1, the extraction trade-off plots for (Llama2-7B, Eval-DU+), and (Llama2-7B, TOFU+) are in Figure 8, 9, 10 respectively;

For the results in Section 4.2, the extraction trade-off plots for (Llama2-7B, Eval-DU+), and (Llama2-7B, TOFU+) are in Figure 11, 12, 13 respectively.

# G  ADDITIONAL EXPERIMENTAL RESULTS

## G.1  EXPERIMENTS WITH THE QWEN3-4B

We additionally conducted all of our designed experiments using a recent model from a different open-source family, **Qwen3-4B**, evaluated on two datasets. The results for Problem 1, Problem 2, Problem 3, Problem 4, and Problem 5 are shown in Figure 14, Figure 15, Figure 16, Figure 17, and

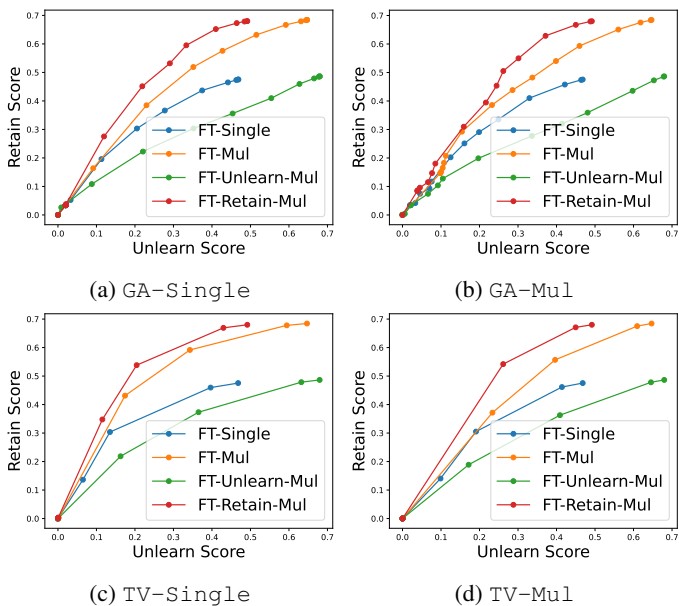

(a) `GA-Single`  (b) `GA-Mul`

(c) `TV-Single`  (d) `TV-Mul`

Figure 8: Vanilla trade-off curves for three choices of unlearning data and two unlearning algorithms on **Eval-DU+** and **Llama2-7B**, when comparing `FT-Mul` and `FT-Single`

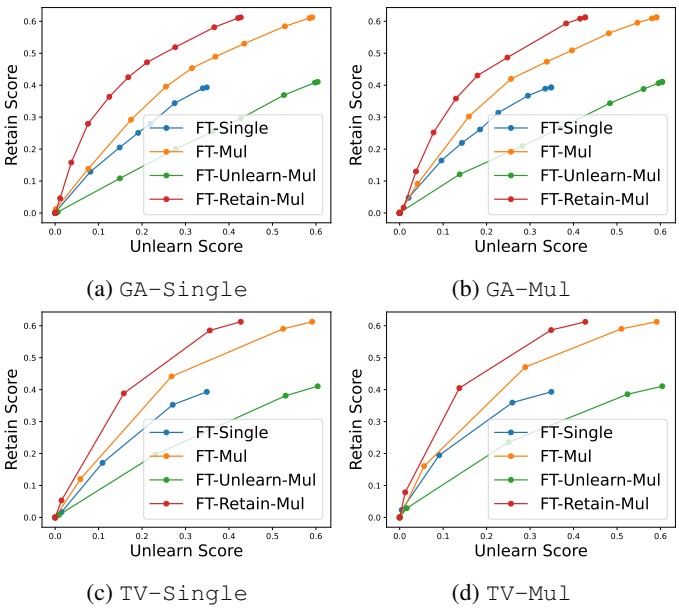

(a) `GA-Single`  (b) `GA-Mul`

(c) `TV-Single`  (d) `TV-Mul`

Figure 9: Vanilla trade-off curves for three choices of unlearning data and two unlearning algorithms on **Eval-DU+** and **Gemma2-2B**, when comparing `FT-Mul` and `FT-Single`

Figure 18, respectively. Across all settings, the results with Qwen3-4B remain consistent with all observations of other models in the main paper used to answer the five problems. **These additional results further strengthen the evidence that our conclusions generalize robustly to a broader range of models.**

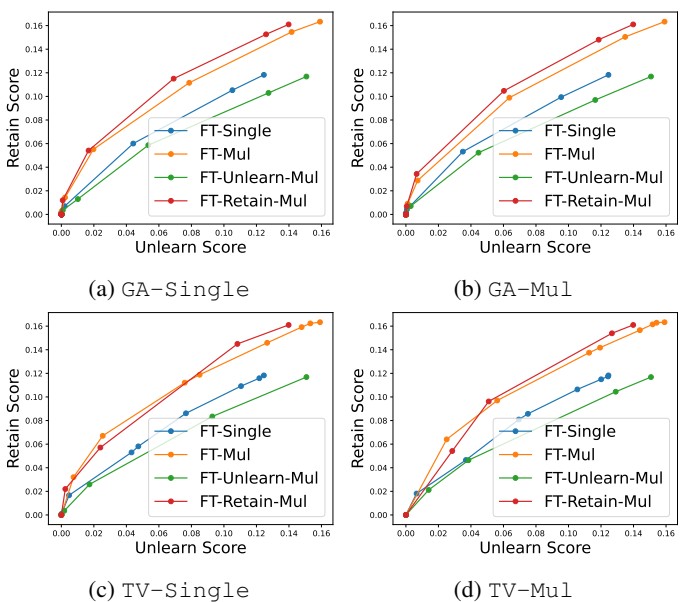

(a) `GA-Single`    (b) `GA-Mul`

(c) `TV-Single`    (d) `TV-Mul`

Figure 10: Vanilla **extraction** trade-off curves for three choices of unlearning data and two unlearning algorithms on **TOFU+** and **Llama2-7B**, when comparing `FT-Mul` and `FT-Single`

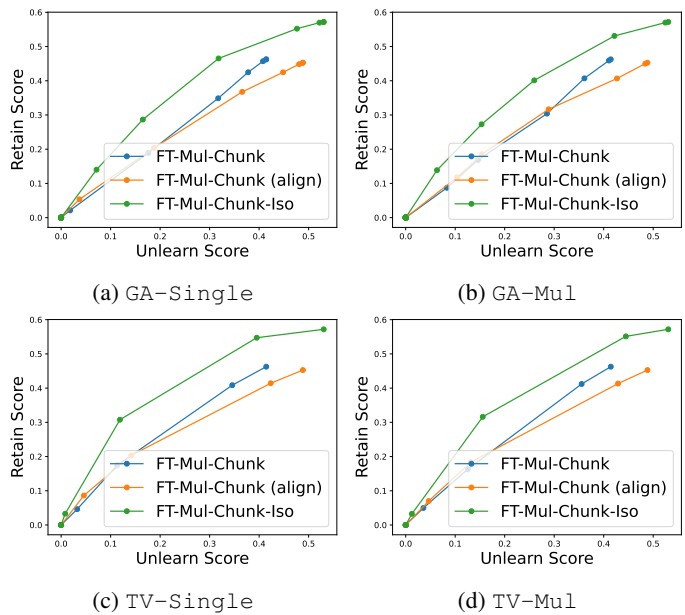

(a) `GA-Single`    (b) `GA-Mul`

(c) `TV-Single`    (d) `TV-Mul`

Figure 11: Vanilla trade-off curves for three choices of unlearning data and two unlearning algorithms on **Eval-DU+** and **Llama2-7B**, when the model is fine-tuned from any text chunks.

## G.2 EVALUATE WITH RETAIN-AWARE UNLEARNING METHODS GRADIENT DESCENT

We additionally evaluate our framework using another retain-aware unlearning method, **Gradient Difference**. This method performs unlearning by ascending the loss on the forget dataset while simultaneously descending the loss on the retain dataset. The retain dataset here consists of all textual descriptions associated with the retain knowledge.

The results for Problem 1, Problem 2, Problem 3, Problem 4, and Problem 5 are shown in Figure 19, Figure 20, Figure 21, Figure 22, and Figure 23, respectively. Across all settings, the results from

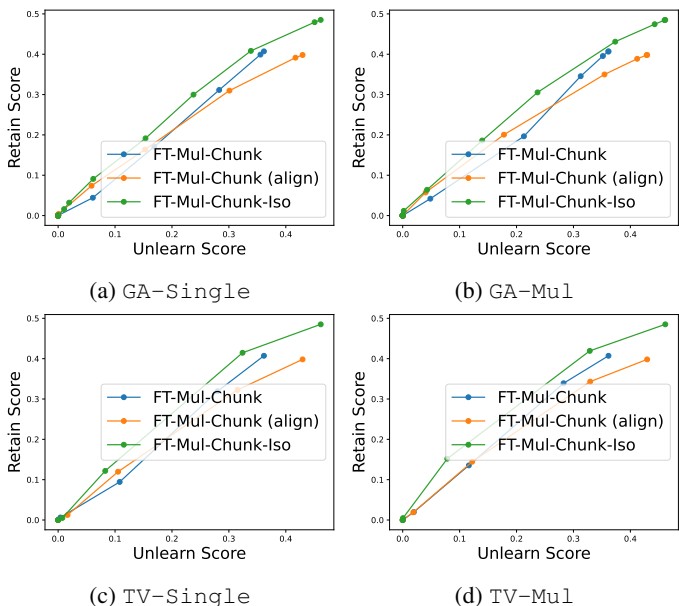

(a) `GA-Single`    (b) `GA-Mul`

(c) `TV-Single`    (d) `TV-Mul`

Figure 12: Vanilla **extraction** trade-off curves for three choices of unlearning data and two unlearning algorithms on **Eval-DU+** and **Gemma2-2b**, when the model is fine-tuned from any text chunks.

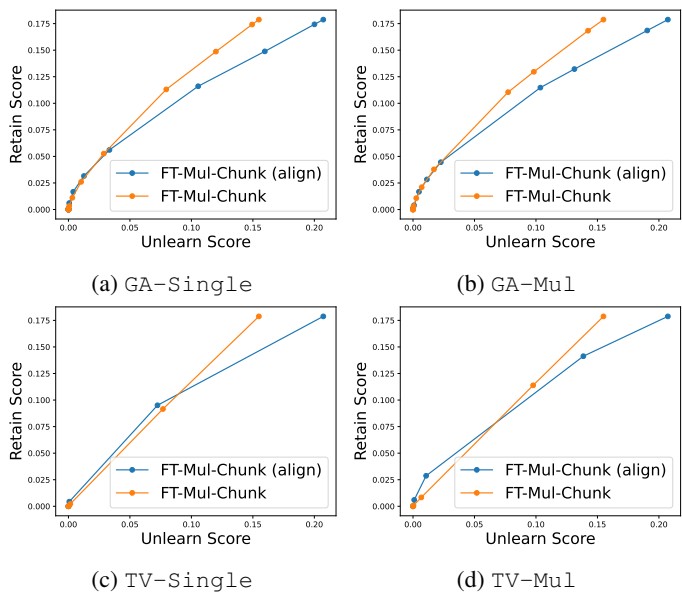

(a) `GA-Single`    (b) `GA-Mul`

(c) `TV-Single`    (d) `TV-Mul`

Figure 13: Vanilla trade-off curves for three choices of unlearning data and two unlearning algorithms on **TOFU+** and **Llama2-7B**, when the model is fine-tuned from any text chunks.

Gradient Difference remain consistent with all observations of other unlearning methods in the main paper used to answer the five problems. **These additional results further demonstrate that that our conclusions are applied to a range of unlearing methods.**

### G.3 EVALUATE WITH THE ADDITIONAL METRIC

We additionally evaluate all unlearning methods using an alternative metric, **Retain@Unlearn** $\tau$, which measures the retain score at the point where the unlearn score has been suppressed to a small target level $\tau$. This metric complements AUC-based evaluations by providing an absolute, threshold-

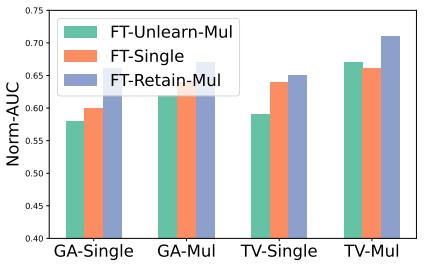
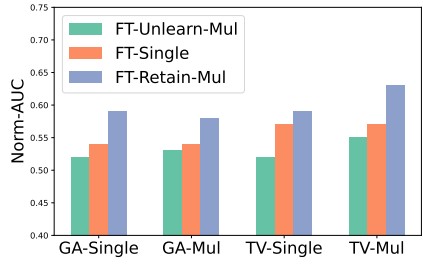

(a) Qwen3-4B, Eval-DU+        (b) Qwen3-4B, TOFU+

Figure 14: Empirical study for Problem 1 on **Qwen3-4B** across two datasets: FT-Single vs FT-Unlearn-Mul vs FT-Retain-Mul.

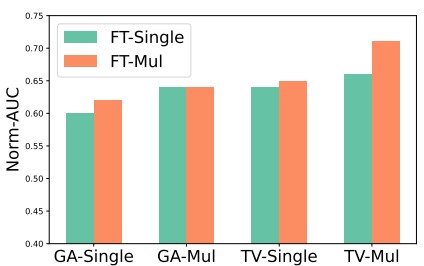
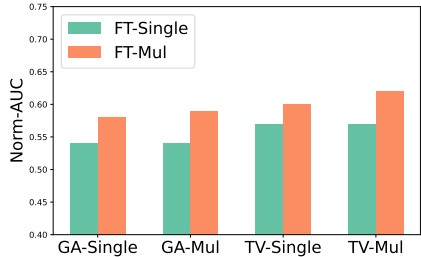

(a) Qwen3-4B, Eval-DU+        (b) Qwen3-4B, TOFU+

Figure 15: Empirical study for Problem 2 on **Qwen3-4B** across two datasets: FT-Single vs FT-Mul.

based view of the unlearning–retaining trade-off. The results for Problem 1, Problem 2, Problem 3, Problem 4, and Problem 5 are presented in Figure 24, Figure 25, Figure 26, Figure 27, and Figure 28, respectively. **Across all experiments, the results under Retain@Unlearn $\tau$ are consistent with our original findings, further validating all observations used to answer the five problems.**

Table 9: This summarize the Norm-AUC of all unlearning results at different setting of unlearning across two datasets and **Qwen3-4B**.

| Model, Dataset | FT Choices | Gradient Ascent | | Task Vector | |
|---|---|---|---|---|---|
| | | GA-Single | GA-Mul | TV-Single | TV-Mul |
| Qwen3-4B, Eval-DU+ | FT-Single | 0.60 | 0.64 | 0.64 | 0.66 |
| | FT-Unlearn-Mul | 0.58 | 0.62 | 0.59 | 0.67 |
| | FT-Retain-Mul | 0.66 | 0.67 | 0.65 | 0.71 |
| | FT-Mul | 0.62 | 0.64 | 0.65 | 0.71 |
| | FT-Chunk-Mul | 0.54 | 0.54 | 0.54 | 0.58 |
| | FT-Chunk-Mul (align) | 0.57 | 0.57 | 0.58 | 0.60 |
| | FT-Chunk-Mul-Iso | 0.58 | 0.58 | 0.56 | 0.59 |
| Qwen3-4B, TOFU+ | FT-Single | 0.54 | 0.54 | 0.57 | 0.57 |
| | FT-Unlearn-Mul | 0.52 | 0.53 | 0.52 | 0.55 |
| | FT-Retain-Mul | 0.59 | 0.58 | 0.59 | 0.63 |
| | FT-Mul | 0.58 | 0.59 | 0.60 | 0.62 |
| | FT-Chunk-Mul | 0.46 | 0.46 | 0.47 | 0.47 |
| | FT-Chunk-Mul (align) | 0.52 | 0.52 | 0.53 | 0.54 |

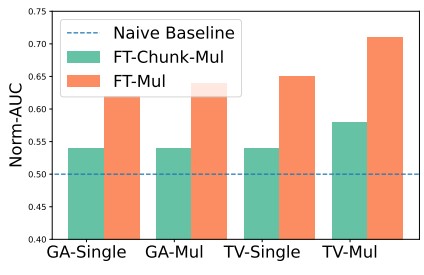
(a) Qwen3-4B, Eval-DU+

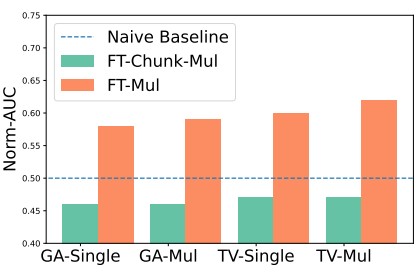
(b) Qwen3-4B, TOFU+

Figure 16: Empirical study for Problem 3 on **Qwen3-4B** across two datasets: FT-Mul-Chunk vs FT-Mul.

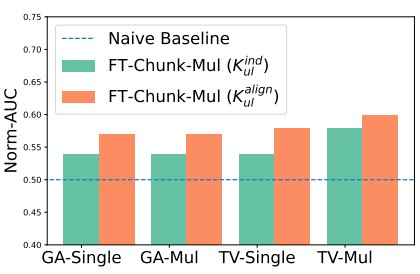
(a) Qwen3-4B, Eval-DU+

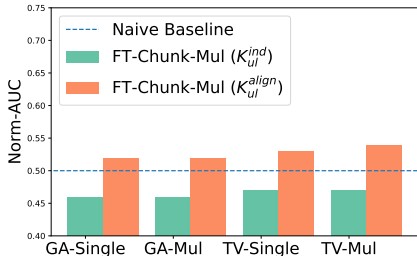
(b) Qwen3-4B, TOFU+

Figure 17: Empirical study for Problem 4 on **Qwen3-4B** across two datasets: FT-Mul-Chunk ($K_{ul}^{align}$) vs FT-Mul-Chunk ($K_{ul}^{ind}$).

Table 10: This table summarizes the Norm-AUC ($\uparrow$) of **Gradient Difference** at different setting of unlearning across two datasets and two models.

| Model, Dataset | FT Choices | Gradient Difference | |
|---|---|---|---|
| | | GD-Single | GD-Mul |
| Llama2-7B, Eval-DU+ | FT-Single | 0.636 | 0.645 |
| | FT-Unlearn-Mul | 0.582 | 0.611 |
| | FT-Retain-Mul | 0.643 | 0.656 |
| | FT-Mul | 0.639 | 0.646 |
| | FT-Chunk-Mul | 0.559 | 0.523 |
| | FT-Chunk-Mul (align) | 0.631 | 0.621 |
| | FT-Chunk-Mul-Iso | 0.622 | 0.648 |
| Llama2-7B, TOFU+ | FT-Single | 0.584 | 0.588 |
| | FT-Unlearn-Mul | 0.577 | 0.581 |
| | FT-Retain-Mul | 0.633 | 0.656 |
| | FT-Mul | 0.636 | 0.644 |
| | FT-Chunk-Mul | 0.472 | 0.476 |
| | FT-Chunk-Mul (align) | 0.647 | 0.733 |

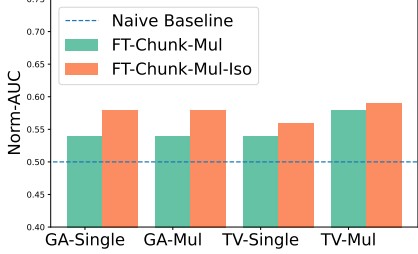
(a) Qwen3-4B, Eval-DU+

Figure 18: Empirical study for Problem 5 on **Qwen3-4B** across two datasets: FT-Mul-Chunk vs FT-Mul-Chunk-ISO.

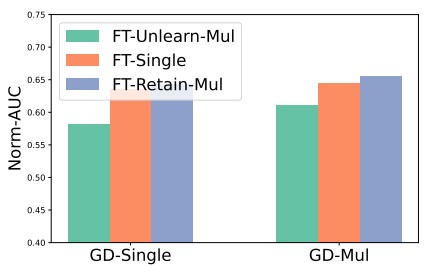 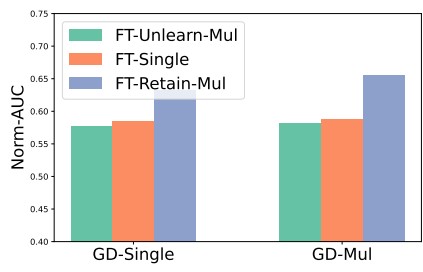

(a) Llama2-7B, Eval-DU+    (b) Llama2-7B, TOFU+

Figure 19: Empirical study for Problem 1 by evaluating **Gradient Difference** (GD-Single, GD-Mul): FT-Single vs FT-Unlearn-Mul vs FT-Retain-Mul.

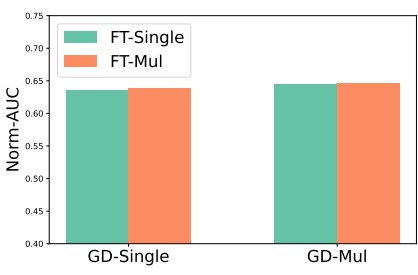 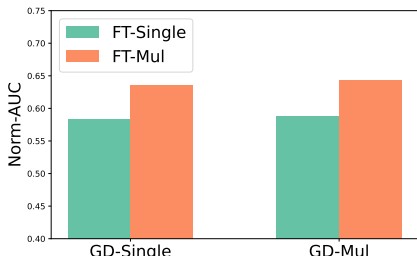

(a) Llama2-7B, Eval-DU+    (b) Llama2-7B, TOFU+

Figure 20: Empirical study for Problem 2 by evaluating **Gradient Difference** (GD-Single, GD-Mul): FT-Single vs FT-Mul.

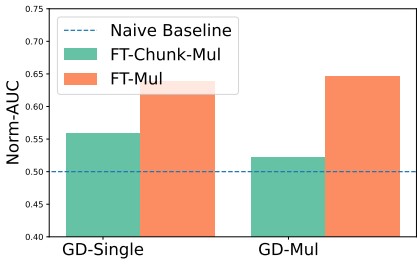 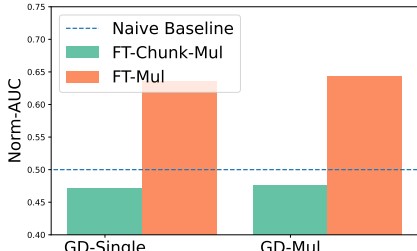

(a) Llama2-7B, Eval-DU+    (b) Llama2-7B, TOFU+

Figure 21: Empirical study for Problem 3 by evaluating **Gradient Difference** (GD-Single, GD-Mul): FT-Mul-Chunk vs FT-Mul.

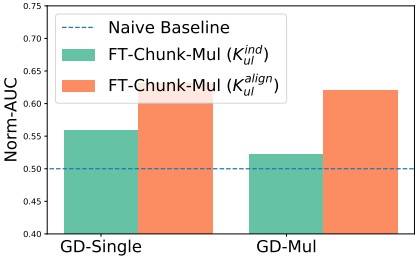 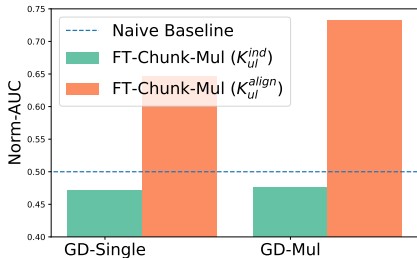

(a) Llama2-7B, Eval-DU+    (b) Llama2-7B, TOFU+

Figure 22: Empirical study for Problem 4 by evaluating **Gradient Difference** (GD-Single, GD-Mul): FT-Mul-Chunk ($K_{ul}^{align}$) vs FT-Mul-Chunk ($K_{ul}^{ind}$).

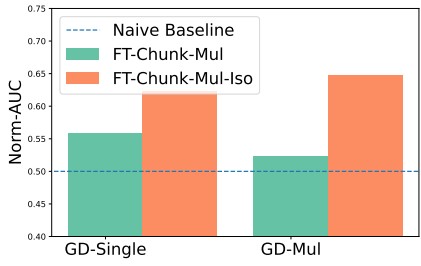

(a) Llama2-7B, Eval-DU+

Figure 23: Empirical study for Problem 5 by evaluating **Gradient Difference** (GD-Single, GD-Mul): FT-Mul-Chunk vs FT-Mul-Chunk-ISO.

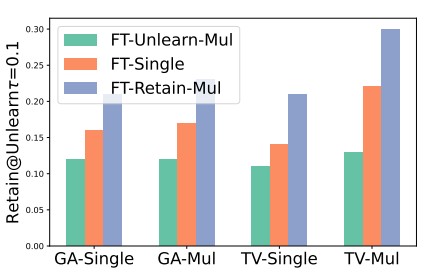
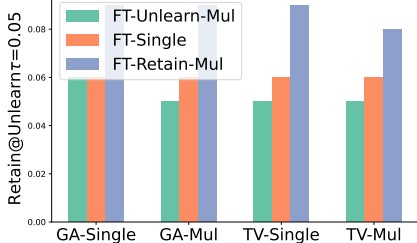

(a) Llama2-7B, Eval-DU+      (b) Llama2-7B, TOFU+

Figure 24: Empirical study for Problem 1 evaluated by **new metric Retain@Unlearn**$\tau$: FT-Single vs FT-Unlearn-Mul vs FT-Retain-Mul.

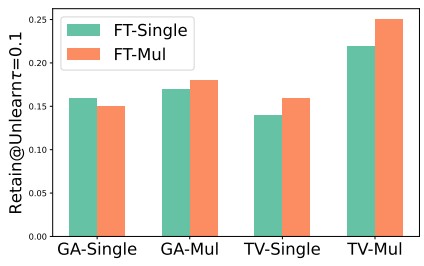
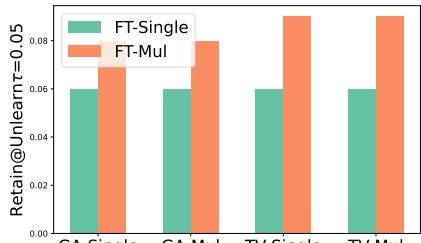

(a) Llama2-7B, Eval-DU+      (b) Llama2-7B, TOFU+

Figure 25: Empirical study for Problem 2 evaluated by **new metric Retain@Unlearn**$\tau$: FT-Single vs FT-Mul.

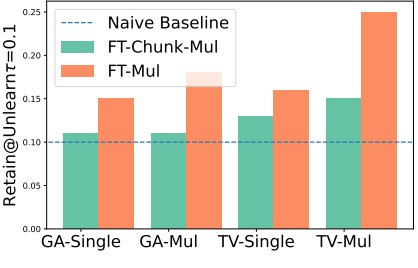
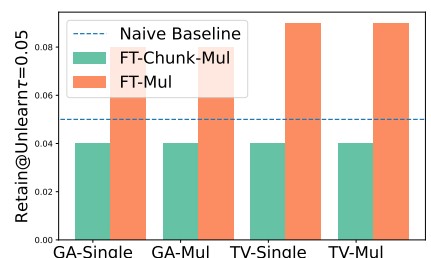

(a) Llama2-7B, Eval-DU+      (b) Llama2-7B, TOFU+

Figure 26: Empirical study for Problem 3 evaluated by **new metric Retain@Unlearn**$\tau$: FT-Mul-Chunk vs FT-Mul.

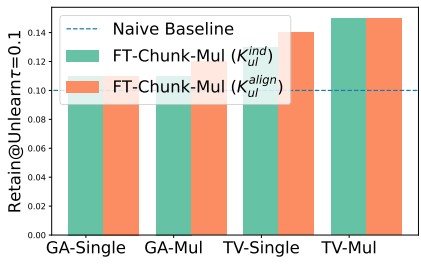 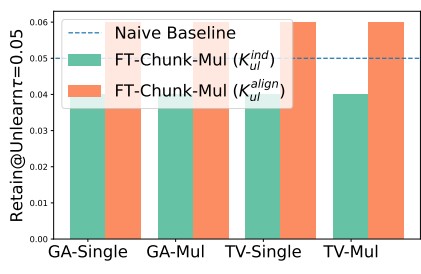

(a) Llama2-7B, Eval-DU+              (b) Llama2-7B, TOFU+

Figure 27: Empirical study for Problem 4 evaluated by **new metric Retain@Unlearn**$\tau$: FT-Mul-Chunk ($K_{ul}^{align}$) vs FT-Mul-Chunk ($K_{ul}^{ind}$).

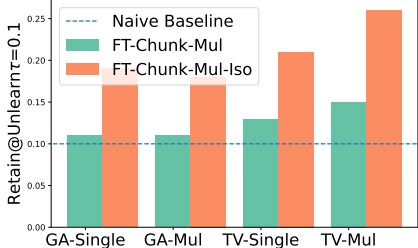

(a) Llama2-7B, Eval-DU+

Figure 28: Empirical study for Problem 5 evaluated by **new metric Retain@Unlearn**$\tau$: FT-Mul-Chunk vs FT-Mul-Chunk-ISO.

Table 11: This summarize the Retain@Unlearn$\tau$ ($\tau = 0.1$ for Eval-DU+ and $\tau = 0.05$ for TOFU+) of all unlearning methods at different setting of training set across two datasets and Llama2-7B.

| Model, Dataset | FT Choices | Gradient Ascent | | Task Vector | |
|---|---|---|---|---|---|
| | | GA-Single | GA-Mul | TV-Single | TV-Mul |
| Llama2-7B, Eval-DU+ | FT-Single | 0.16 | 0.17 | 0.14 | 0.22 |
| | FT-Unlearn-Mul | 0.12 | 0.12 | 0.11 | 0.13 |
| | FT-Retain-Mul | 0.21 | 0.23 | 0.21 | 0.30 |
| | FT-Mul | 0.15 | 0.18 | 0.16 | 0.25 |
| | FT-Chunk-Mul | 0.11 | 0.11 | 0.13 | 0.15 |
| | FT-Chunk-Mul (align) | 0.11 | 0.12 | 0.14 | 0.15 |
| | FT-Chunk-Mul-Iso | 0.19 | 0.18 | 0.21 | 0.26 |
| Llama2-7B, TOFU+ | FT-Single | 0.06 | 0.06 | 0.06 | 0.06 |
| | FT-Unlearn-Mul | 0.06 | 0.05 | 0.05 | 0.05 |
| | FT-Retain-Mul | 0.09 | 0.09 | 0.09 | 0.08 |
| | FT-Mul | 0.08 | 0.08 | 0.09 | 0.09 |
| | FT-Chunk-Mul | 0.04 | 0.04 | 0.04 | 0.04 |
| | FT-Chunk-Mul (align) | 0.06 | 0.06 | 0.06 | 0.06 |

