# OpenReview forum: "Learning-Time Encoding Shapes Unlearning in LLMs"
_ICLR.cc/2026/Conference — ICLR 2026 Poster_

### Official Review · Reviewer_PLQ7 · 2025-10-27

**Soundness:** 1
**Presentation:** 3
**Contribution:** 2
**Rating:** 2
**Confidence:** 4

**Summary:**

This paper addresses the problem of knowledge unlearning in large language models (LLMs). While prior work has primarily focused on developing new methodological approaches to improve unlearning performance on public benchmarks, this paper instead investigates how the way target knowledge is encoded in the training data affects the effectiveness of unlearning. To this end, the authors consider two main experimental settings: (1) examining the effect of text paraphrasing on unlearning, and (2) examining the effect of text chunk composition on unlearning. Through a series of controlled experiments, the paper observes that models trained with paraphrased data exhibit more effective unlearning, and that isolating forget knowledge within text chunks leads to more efficient forgetting.

**Strengths:**

* This paper approaches the unlearning problem from a perspective, analyzing how the learning-time encoding (the way target knowledge is represented and learned during training) affects the effectiveness of unlearning. The framing that "how and what is learned determines how well it can be forgotten" clearly distinguishes this work from prior studies focused solely on algorithmic improvements. This is a valid and valuable perspective for deepening our understanding of knowledge unlearning in LLMs.
* The paper systematically separates and investigates three factors, text paraphrasing, chunk entanglement, and sentence isolation, to examine how each influences the difficulty of unlearning. This stepwise, controlled design provides a meaningful experimental setup for exploring the structural causes of unlearning difficulty.

**Weaknesses:**

* In each experimental setting (e.g., FT-Single vs FT-Unlearn-Mul), the initial learning strength of the forget and retain knowledge differs. Models trained with paraphrased data tend to encode the same facts more strongly, resulting in higher initial scores and making unlearning appear more difficult. The paper attempts to account for this difference using the Norm-AUC metric, but this measure has a structural limitation: models with higher initial scores may be disadvantaged in relative evaluation, since the same absolute decrease in score yields a smaller relative change. Consequently, Norm-AUC does not fully normalize for differences in learning intensity. It would be more appropriate to analyze unlearning difficulty using models that are fine-tuned to have comparable initial forget/retain scores.
* The results for Problem 3 and Problem 4 appear relatively self-evident given the experimental setup. When the forget and retain sets coexist within the same chunk, unlearning naturally fails, and when they are fully separated by chunk boundaries, performance improves as gradient interference is removed. These experiments therefore confirm rather than extend what is already understood about representation entanglement in unlearning.
* The improvement observed in Problem 5 also lacks a sufficiently clear explanation. The only difference in this setting is that facts are arranged as independent sentences rather than connected text, yet the paper provides no theoretical or quantitative justification for why this should make unlearning more effective. While the empirical trend is interesting, the causal interpretation of this result remains underdeveloped.
* The paper evaluates only two unlearning algorithms, Gradient Ascent (GA) and Task Vector (TV). Although these represent the optimization-based and representation-editing paradigms, respectively, this scope is too narrow to support general conclusions about unlearning mechanisms. It would be beneficial to include retain-aware or preference-based algorithms (e.g., Gradient Difference, Direct Preference Optimization, etc.) that explicitly leverage retain data during optimization, as such methods may exhibit different behavior with respect to the proposed encoding effects.
* Minor typos
  * Line 125: paraphased -> paraphrased
  * Line 290: only three combinations are listed

**Questions:**

* In Section 2.3, the authors justify focusing on the fine-tuning stage for studying unlearning. However, fine-tuning typically involves a much smaller and more carefully curated dataset, often with significant manual filtering. In contrast, the pretraining stage usually carries a much higher risk of including sensitive or private user data. Could the authors elaborate on why fine-tuning is considered a more realistic or representative setting for privacy-sensitive unlearning?
* As a learning-time strategy for improving the post-hoc efficiency of unlearning, the paper proposes separating knowledge during training. However, in the Problem 5 setting, where each fact is trained as an independent sentence, might this strategy introduce side effects such as degraded fluency, loss of contextual coherence, or weaker entity-level reasoning in generated text? Have the authors observed or considered such trade-offs?

---

> ### Author Response · Authors · 2025-11-29
>
> # Summary of Revisions and Responses
>
> We thank the reviewer for acknowledging that our study is in a valid and valuable aspect and our experimental set-up is systematic and meaningful. The reviewer suggested several aspects to strengthen the generalizability of our findings. In response, we believe we well addressed the reviewer's concerns by strengthening both our analyses and empirical evaluations. Here is the summary of our responses:
> 1. (W1) To address concerns about differing initial knowledge strength, we clarified our training-configuration rationale and introduced a new absolute evaluation metric, Retain@Unlearn τ, whose results (App. G.3) consistently support all conclusions.
> 2. (W2, W3) We expanded our explanation of Problems 3–5 by distinguishing textual entanglement from standard gradient interference, demonstrating why the observed effects are non-trivial and not implied by prior intuition.
> 3. (W4) We added new experiments using the retain-aware Gradient Difference method (App. G.2), confirming that our observations hold across additional unlearning paradigms.
> 4. (Q1) We clarified why fine-tuning is the most realistic setting for privacy-sensitive unlearning in current practice and aligned with established benchmarks (TOFU, MUSE).
> 5. (Q2) We also discussed potential trade-offs in Problem 5, emphasizing that single-sentence isolation is used only as a controlled probe and not as a recommended practical recipe.
> 6. (W5) Minor typos have been corrected.
>
> Please check the detailed responses as follows.
>
> ## Evaluation metrics & training configuration calibration (W1).
> Thank you for raising this thoughtful point. We appreciate the concern regarding differences in initial knowledge strength across training configurations, and we address it from both an experimental-design and evaluation-metric perspective.
>
> First, regarding the choice of training configurations: **our goal is to simulate realistic fine-tuning practices**. For each setting, we fine-tune until the model begins to converge on the corresponding training set, which mirrors how practitioners would actually train models with different data volumes or paraphrasing strategies. While this does induce different initial forget/retain scores, we intentionally preserve these differences because they reflect the true consequences of dataset choice. In realistic deployments, unlearning is applied to models whose retained and forgotten knowledge may naturally be learned with different intensities; **forcing the initial scores to be artificially matched would remove an important aspect of practice.** Importantly, unlearning does not require forgetting and retaining knowledge to start at the same level, nor would such alignment occur naturally.
>
> Second, regarding the Norm-AUC metric: we agree that Norm-AUC captures relative score reduction, and that this may disadvantage models with higher initial scores. Norm-AUC was designed to provide a normalized view across diverse settings, but we acknowledge its inherent imperfectness as a purely percentage-based measure. **To address this, we incorporate an additional absolute evaluation metric: Retain@Unlearn $\tau$.** This metric asks: when the forgotten knowledge is suppressed to a small target level $\tau$, what retain-knowledge score does the model preserve? Retain@Unlearn$\tau$ complements Norm-AUC by directly comparing unlearning performance at matched unlearned-knowledge endpoints rather than relying on initial score normalization.
> We have added the Retain@Unlearn$\tau$ analysis to the Appendix G.3 (Figure 24,25,26,27,28). **Across all experiments, the results under Retain@Unlearn $\tau$ are consistent with our original findings, further validating all observations used to answer the five problems.** We appreciate that the reviewer proposes this aspect.

---

> ### Author Response · Authors · 2025-11-29
>
> ## The understanding for Problem 3,4 (W2)
> We appreciate the reviewer’s observation and would like to clarify that the contribution of Problems 3 and 4 goes beyond reiterating the general notion of “entanglement” previously discussed in the literature. **Prior work primarily focuses on gradient-level interference. In contrast, our findings reveal a distinct and previously underexplored phenomenon: textual entanglement**, i.e., the structural and lexical intermixing of forget and retain content within the training text itself.
> Importantly, the empirical results we observe cannot be fully explained by gradient interference alone. If gradient interference were the complete explanation, then, as the reviewer hypothesizes, any setting where forget and retain facts co-occur in the same chunk should lead to unlearning failure. However, this is not consistent with our findings. In Problem 5, both FT-Mul-Chunk and FT-Mul-Chunk-ISO place forget and retain sets within the same chunk, yet they exhibit notably different unlearning outcomes. This discrepancy demonstrates that gradient co-occurrence is not sufficient to explain the behavior.
> Instead, the critical factor is how the forget and retain information co-occur at the level of textual description.
>     - In FT-Mul-Chunk, the forget and retain facts share vocabulary, phrasing, and narrative structure. The description of the forget knowledge is intertwined with retain information across multiple parts of the paragraph.
>     - In FT-Mul-Chunk-ISO, the forget and retain facts appear in the same chunk but are lexically and semantically isolated, with minimal shared wording or interleaving.
>
> This difference in textual entanglement leads to different internal representations and ultimately different unlearning difficulty, even though gradient-level co-occurrence is identical.
>
> Thus, **Problems 3–5 collectively highlight a new and non-obvious insight: the structure and lexical organization of the training text, not merely chunk-level co-occurrence, plays a crucial role in whether unlearning succeeds.** This extends the understanding of entanglement in unlearning and provides a more fine-grained explanation of empirical behavior, offering guidance for both dataset construction and unlearning algorithm design.
>
> ## The understanding for Problem 5 (W3)
> Our response for weakness 2 above has covered the explanation for the observation in Problem 5 as well. We will add these discussions in our revision. Please let us know if you have any further questions about our explanation.
>
> ## Additional evaluation on retain-aware unlearning (W4)
> Thanks for pointing out the retain-aware unlearning algorithm. We additionally experimented with Gradient Difference that explicitly leverages retain data for our five proposed problems. The results are presented in Appendix G.2 (Figure 19, 20, 21, 22, 23). We find that all results of Gradient Difference still support our answers to the study of five problems. Notably, even with the retained data, the unlearning from FT-Mul-Chunk is very ineffective. We appreciate the suggestion of an additional type of unlearning algorithm, which enhances the generability of our conclusion.

---

> ### Author Response · Authors · 2025-11-29
>
> ## The Usecase of privacy-preserving unlearning in fine-tuning (Q1)
> We agree that unlearning at the pretraining stage is an important and challenging long-term direction. However, this does not diminish the relevance of studying unlearning at the fine-tuning stage. In practice, fine-tuning is often the most realistic and representative setting where privacy-sensitive unlearning is actually required.
>
> First, **fine-tuning is precisely the stage where organizations inject their own proprietary or user-identifiable data into a model.** Examples include customer logs for support bots, patient summaries for clinical assistants, internal documents for enterprise copilots, social media posts and interactions for the assistant agents. Because this data originates directly from identifiable users or regulated sources, it falls squarely under privacy laws such as GDPR, CCPA, FERPA, and HIPAA. As a result, the vast majority of real-world unlearning requests arise from fine-tuning data, not from the anonymous and untraceable pretraining corpus.
>
> Second, **our choice aligns with established research practice.** The most widely used LLM unlearning benchmarks—including TOFU [1] and MUSE [2]—are explicitly designed for evaluating unlearning in fine-tuned models. Much of the literature on LLM unlearning, model editing, and knowledge removal therefore adopts this setting as the standard experimental paradigm.
> For these reasons, we believe fine-tuning provides a realistic, actionable, and representative environment for studying privacy-sensitive unlearning today, while acknowledging that pretraining-stage unlearning remains an important open challenge.
>
> [1] Maini, P., Feng, Z., Schwarzschild, A., Lipton, Z. C., & Kolter, J. Z. TOFU: A Task of Fictitious Unlearning for LLMs. In First Conference on Language Modeling.
>
> [2] Shi, W., Lee, J., Huang, Y., Malladi, S., Zhao, J., Holtzman, A., ... & Zhang, C. MUSE: Machine Unlearning Six-Way Evaluation for Language Models. In The Thirteenth International Conference on Learning Representations.
>
> ## Trade-off in separating knowledge (Q2)
> We would like to clarify that we are not intended to recommend full-sentence isolation as a universal training practice. Rather, the purpose of Problem 5 is to study the mechanisms that make unlearning more or less feasible. **The single-sentence training condition serves as an extreme controlled scenario that allows us to disentangle textual interference effects.** Our findings do not suggest that practitioners should adopt single-sentence training in real deployments. **Instead, a reasonably promising extension is to isolate only the boundary between retain and unlearn content**, which may substantially reduce any impact on fluency while still leveraging the same mechanism to improve unlearning performance. We consider the practical exploration of such partial isolation strategies as future work and will clarify this trade-off more explicitly in the revision.

---

### Official Review · Reviewer_bmBj · 2025-10-30

**Soundness:** 2
**Presentation:** 2
**Contribution:** 2
**Rating:** 6
**Confidence:** 3

**Summary:**

The paper investigates how the way knowledge is encoded during fine-tuning impacts the effectiveness of post-hoc knowledge unlearning in LLMs. The authors argue that how knowledge is represented (shaped) in the training corpus significantly influences how difficult it is to later remove.
The authors conduct a rigorous, controlled empirical study by fine-tuning two LLMs (Llama2-7B and Gemma2-2B) on two extended benchmarks (Eval-DU+ and TOFU+) which use synthetic, fictitious knowledge to avoid pre-training contamination. They systematically test two main encoding factors:
- Paraphrasing: They compare models trained on single vs. multiple paraphrased descriptions of facts.
- Text Entanglement: They analyze unlearning when facts-to-be-forgotten ("forget set") are embedded in the same text chunk (e.g., a paragraph) as facts-to-be-kept ("retain set").

Their results consistently show that unlearning is harder when the forget set facts were paraphrased, but easier when the retain set facts were paraphrased.  Critically, the study also finds that unlearning individual facts is exceptionally difficult when forget and retain facts are entangled in the same training text chunk.

**Strengths:**

- The paper explores a novel aspect on data shapes for unlearning in LLMs. Most unlearning research focuses on post-hoc algorithms. This work provides a new and important perspective by showing that data curation strategies during fine-tuning are a critical and overlooked factor.
- The findings are clear and consistent across two different model families (Llama2, Gemma2) , two different datasets (Eval-DU+ and TOFU+), and two representative unlearning algorithms (Gradient Ascent and Task Vectors).

**Weaknesses:**

- Paraphrasing vs. Frequency: The effect of using multiple paraphrases (e.g., in FT-Unlearn-Mul) is closely related to simply increasing the frequency of the fact in the training data. The paper argues this encourages "structured" representations and distinguishes itself from related work on frequency, but it doesn't empirically disentangle the effect of the paraphrasing from the frequency.
- No real-world dataset: The use of synthetic data is a key strength for experimental control, but also a potential weakness. Real-world corpora are far messy, and the entanglement of facts is likely more complex than the binary "entangled chunk" (FT-Mul-Chunk) or "isolated sentences" (FT-Mul-Chunk-Iso) settings explored.
- Missing quality checking for the data augmentation (e.g., paraphrasing, separating) via GPT4o.
- In the paper, the author mentions four combinations of models and datasets. However, only three of them are shown, Gemma2 with TOFU+ is missing.

**Questions:**

- While you acknowledge the limitation of focusing on fine-tuning, do you have any hypotheses on how these findings might translate to the pre-training regime? Knowledge is both heavily paraphrased and heavily entangled during pre-training. Does this imply that unlearning pre-trained knowledge will always be as difficult as your FT-Mul-Chunk scenario, unless the fact is extremely rare?
- Can you explicitly explain the baselines represents a random-chance unlearning in Figure 4,5,6?
- Is FT-Mul-Chunk-Iso part of FT-Mul-Chunk?
- Could you please provide new results based on LLama3 and Gemma3?

---

> ### Author Response · Authors · 2025-11-29
>
> # Summary of Revisions and Responses
> We thank the reviewer for acknowledging our work provides a new and important perspective and shows consistent results across a comprehensive setttings. In response, we conducted new analyses and added clarifications. Specifically:
> 1. (W1) We add the analysis where we disentangled paraphrasing effects from frequency by adding new “frequency-only” controls, confirming paraphrasing, not repetition, drives the observed behavior in our experiments.
> 2. (W2, Q1) We added a discussion on how our insights may extend to real-world and pretraining settings.
> 3. (W3, Q2) We clarified dataset quality checks and baseline interpretations.
> 4. (Q3) We explained the distinction between FT-Mul-Chunk and FT-Mul-Chunk-Iso.
> 5. (Q4) We expanded model coverage by evaluating an additional model family (Qwen3-4B), which replicates all key findings.
>
> Please check the detailed responses as follows.
>
> ## Frequency vs Paraphrasing (W1)
> Thanks for raising this angle! We construct two variants `FT-Unlearn-Mul (Freq)` and `FT-Retain-Mul  (Freq)` for `FT-Unlearn-Mul` and `FT-Retain-Mul` respectively, where in the variants of `(Freq)` we replace “paraphrasing” by “repeating” when constructing the dataset. The following are the AUC for GA-Mul on Eval-DU+ dataset:
>
> |                               | GA-Mul |
> |-------------------------------|--------|
> | FT-Single                     | 0.628  |
> | FT-Unlearn-Mul (Freq)    | 0.596  |
> | FT-Unlearn-Mul | 0.576  |
> | FT-Retain-Mul (Freq)     | 0.621  |
> | FT-Retain-Mu  | 0.639  |
>
> From the results, we could find that `FT-Retain-Mul (Freq)` is even worse than `FT-Single`, while
> `FT-Retain-Mul` with paraphrasing is consistently better than `FT-Single` shown in our paper Figure 2. **This divergence demonstrates that our results cannot be attributed to the frequency but the paraphrasing.**
>
> ## No real-world dataset (W2) & Insights to the pre-trained knowledge unlearning (Q1)
> We agree that the real-world dataset such as the pre-train corpus can be much more complicated. Overall, we believe our findings still provide the intuition to explain some unlearning behaviors when the corpus is more complicated, though we also agree that it is important to verifying these hypotheses for complicated corpus empirically in the future work. We will add the related discussion in revision. For more details, **please check our response “Insights to the pre-trained knowledge unlearning” in the reply to Reviewer Av7s.**
>
> ## Missing quality checking (W3)
> We conducted the manual checking for the quality to our best and will add this detail to the paragraph describing the dataset.
>
> ## Baselines in Figure 4,5,6 (Q2)
> he baseline corresponds to an area under the curve (AUC) of 0.5, which represents a “random-chance” unlearning behavior. In this case, the retain and unlearn scores decrease at approximately the same rate as the unlearning strength increases, producing an almost straight, diagonal curve (e.g., the FT-Mul-Chunk curves in Figure 11). In contrast, an effective unlearning method will have an AUC greater than 0.5, resulting in a concave curve where the unlearn score drops faster than the retain score (e.g., the FT-Mul curve in Figure 8). We will add this explanation in revision.
>
> ## FT-Mul-Chunk-Iso vs FT-Mul-Chunk (Q3)
> Sorry about this confusion! The FT-Mul-Chunk should be the case where the forget and retain information not only co-occur at the level of textual description, but also **they share vocabulary, phrasing, and narrative structure**. The description of the forget knowledge is intertwined with retain information across multiple parts of the paragraph. In FT-Mul-Chunk-ISO, the forget and retain facts appear in the same chunk but are lexically isolated, with minimal shared wording or interleaving.
>
> ## Results on more models (W4, Q4)
> Thank you for the suggestion to evaluate additional models to further strengthen the robustness of our findings. **In response, we have conducted experiments using a recent model from a different open-source family Qwen3-4B.** We replicated all experiments across all problem settings in our paper on two datasets, and the corresponding results are now included in the Appendix G.1 (Figure 14,15,16,17,18). The observations are consistent with our original conclusions, demonstrating that our findings hold across model families. We will integrate these new results into the main paper in the revised version.

---

### Official Review · Reviewer_Av7s · 2025-10-31

**Soundness:** 3
**Presentation:** 4
**Contribution:** 3
**Rating:** 6
**Confidence:** 4

**Summary:**

This paper investigates the effect of shaping of factual knowledge during fine-tuning phase on the effectiveness of knowledge unlearning in LLMs. The analysis reveals.that the knowledge demonstrated in various paraphrased ways are harder to unlearn, while unlearning is more efficient when knowledge is presented with various paraphrased formats overall. Moreover, the paper shows a novel insight that knowledge that is presented inside a single chunk of bath data are harder to unlearn while retaining the other knowledge inside the same chunk.

**Strengths:**

(S1) This work is well motivated, and the core question is important and novel, while not trivial.

(S2) I commend the authors for the clear organization of research questions, appropriate experimental designs, and well-organized writing.

**Weaknesses:**

(W1) **Limited mechanistic understanding**: While the intuition that presenting factual knowledge in multiple paraphrased format leads to more structured representation is compelling and aligns with experimental results, the analysis relies on the observation of knowledge unlearning success, and the understanding on the mechanism governing this behavior is limited. For example:
- Is there any difference in the distribution of the update vector or knowledge circuits [1], that is computed for unlearning single knowledge and paraphrased knowledge, respectively?
- The improved effectiveness of unlearning upon paraphrasing both knowledge types (regarding RQ2) might be attributed to the increased training steps rather than the structures of representation induced by paraphrase, as the size of the fine-tuning dataset is increased. How can we remove this possibility?

(W2) **Scope of the experiment**: While the authors have well justified the experimental setup of unlearning factual knowledge that is first encountered during fine-tuning phase, previous work ([2,3]) have demonstrated that the knowledge obtained during pretraining and fine-tuning phase may be encoded in a different way. This somewhat limits the scope of this work’s contribution. Could you share your thoughts on the applicability of the insights provided in this work to the unlearning of the knowledge acquired during pretraining?



[1] https://arxiv.org/abs/2405.17969
[2] https://arxiv.org/abs/2405.05904
[3] https://arxiv.org/abs/2503.21676

**Questions:**

Please see the questions above.

---

> ### Author Response · Authors · 2025-11-29
>
> # Summary of Revisions and Responses
> We thank the reviewer for acknowledging that our work is well motivated, important, novel, and non-trivial and our paper is well-written. In response:
> 1. (W1-1) For mechanism-level understanding, we acknowledge that analyzing update vectors/knowledge circuits is valuable but beyond scope, and we clarify this as future work.
> 2. (W1-2) We also verified that training-step differences do not drive our results: FT-Single trained with 3× more epochs shows unchanged unlearning behavior.
> 2. (W2) Regarding applicability to pretraining, we discuss how our findings on paraphrasing and textual entanglement may extend to pretraining while recognizing the added complexity of pretraining corpora and noting that empirical validation remains future work.
>
> These clarifications strengthen the scope and interpretation of our contributions.
>
> ## Mechanism-level understanding (W1-1)
> Thank you for highlighting this insightful direction regarding the underlying mechanisms behind our empirical findings. We agree that analyzing differences in update vectors or knowledge circuits between single-form and paraphrased knowledge would offer a deeper mechanistic understanding. However, we also view this analysis as non-trivial: even if such differences are observed, interpreting how they translate into unlearning performance requires careful investigation that goes beyond the scope of our current study. We believe this represents a valuable and promising avenue for future work, and we will add a discussion of this direction in our revision.
>
> ## Training Configuration Control (W1-2)
>  We would like to first clarify our choice of training configurations: **our goal is to simulate realistic fine-tuning practices.** For each setting, we fine-tune until the model begins to converge on the corresponding training set, which mirrors how practitioners would actually train models with different data volumes or paraphrasing strategies. Notice that while this does induce differences for example the initial forget/retain scores, we intentionally preserve these differences because they reflect the true consequences of dataset choice. In realistic deployments, unlearning is applied to models whose retained and forgotten knowledge may naturally be learned with different intensities; **forcing the other set-up to be artificially matched would remove an important aspect of practice**.
>
> Regarding the specific concern about “increased training steps,” our setup selects the number of epochs such that the model has already converged on the training data. This means that increasing the number of epochs for FT-Single, e.g., to match the total number of training steps used in the FT-Mul setting, should not substantially alter the learned weights, since the model is already near convergence. Empirically, we verified this intuition: when we increased the FT-Single training from 5 epochs (used in the main experiments) to 15 epochs (to match FT-Mul), the resulting unlearning performance remained mostly unchanged. **This observation supports our claim that differences in training steps are not driving the effects we study.**

---

> ### Author Response · Authors · 2025-11-29
>
> ## Insights to the pre-trained knowledge unlearning (W2)
> Thank you for raising this thoughtful question, which helps clarify the scope of our work and discuss potential implications beyond it. While our experiments focus on knowledge first introduced during fine-tuning, we believe several of our insights can meaningfully transfer to the pretraining setting. Below we elaborate on how our findings may extend to the two main factors  (text paraphrasing and text chunk structure) studied in our work while acknowledging the additional complexity of pretraining corpora.
>
> **Text paraphrasing.** Unlike fine-tuning data, pretraining corpora are highly heterogeneous: some domains (e.g., domain A) are represented with rich paraphrasing diversity, while others (e.g., domain B) appear far less frequently. Under this disparity, a reasonable transfer hypothesis is: unlearning a specific fact while retaining related knowledge may be easier in domains with high paraphrastic coverage (domain A), because the knowledge space in this domain can be more structured for supporting the unlearning edit. Conversely, in sparsely represented domains (domain B), unlearning may be more brittle. This aligns conceptually with our findings that textual paraphrasing affects unlearning–retaining trade-offs.
>
> **Text chunk structure.** In pretraining, a single fact can appear across many different chunks and co-occur with many different pieces of knowledge. A forget target may be textually entangled with many more surrounding pieces, suggesting that unlearning it may cause broader forgetting.
>
> Overall, we believe the mechanisms highlighted in this work remain relevant to pretraining-time unlearning. In the revised discussion section, we will explicitly include this broader perspective. In the meantime, we will acknowledge the distinctions of knowledge learning between the pre-training and fine-tuning highlighted by prior literature (e.g. [2,3] mentioned in the review), clarifying that it is still important verifying these hypotheses empirically in the future work.

---

### Meta-Review · Area_Chair_ikHc · 2026-01-06

**Summary:**

In the original reviews, the reviewers' concerns centered on several key aspects: the lack of mechanistic understanding behind the observed unlearning effects (Av7s, PLQ7), the potential conflation of paraphrasing with simple frequency increases in training data (bmBj), the limited scope of experiments focusing solely on fine-tuning rather than pre-training knowledge (Av7s, bmBj), and issues with evaluation metrics and experimental design, such as differing initial knowledge strengths and the use of synthetic datasets (PLQ7).

The authors' rebuttal has effectively addressed most of the concerns, so I recommend Accept.

**Reviewer Concerns:**

The authors' rebuttal effectively addressed most of the concerns: they disentangled paraphrasing from frequency through controlled experiments, added experiments with an additional model (Qwen3-4B) and a retain-aware unlearning method (Gradient Difference) to improve generalizability, clarified the distinction between textual entanglement and gradient interference, and introduced a new absolute metric to complement Norm-AUC.

**Reviewer Scores:**

Reviewer Av7s (Score: 6), who appreciated the work's motivation and clarity, would likely maintain the positive score, due to the authors' thorough responses and additional controls, though mechanistic limitations remain.

Reviewer bmBj (Score: 6), concerned about frequency vs. paraphrasing and model coverage, would likely increase the score given the strong new experiments disentangling these factors and expanded model evaluation.

Reviewer PLQ7 (Score: 2), who was more critical and recommended rejection, would likely increase the score due to the added metrics and algorithmic validation, and enriched explanations on experimental design.

---

### Decision · Program_Chairs · 2026-01-26

Accept (Poster)